# High-resolution (1-km) all-sky net radiation over Europe enabled by the merging of land surface temperature retrievals from geostationary and polar-orbiting satellites.

Dominik Rains[1], Isabel Trigo[2], Emanuel Dutra[2], Sofia Ermida[2], Darren Ghent[3], Petra Hulsman[1], Jose Gómez-Dans[4], and Diego G. Miralles[1]

[1]Hydro-Climate Extremes Lab (H-CEL), Ghent University, Ghent, Belgium
[2]Instituto Português do Mar e a Atmosfera, Lisboa, Portugal
[3]University of Leicester, Space Research Centre, Leicester, United Kingdom
[4]King's College London, Bush House, London, United Kingdom

**Correspondence:** Dominik Rains (dominik.rains@airbus.com)

**Abstract.** Surface Net Radiation (SNR) is a vital input for many land surface and hydrological models. However, current remote sensing datasets of SNR come mostly at coarse resolutions or have large gaps due to cloud-cover that hinder their use as input in models. Here, we present a downscaled and continuous daily SNR product across Europe for 2018–2019. Longwave outgoing radiation is computed from a merged land surface temperature (LST) product in combination with Meteosat Second Generation emissivity data. The merged LST product is based on all-sky LST retrievals from the Spinning Enhanced Visible and InfraRed Imager (SEVIRI) onboard the geostationary Meteosat Second Generation (MSG) satellite, and clear-sky LST retrievals from the Sea and Land Surface Temperature Radiometer (SLSTR) onboard the polar-orbiting Sentinel 3A satellite. This approach makes use of the medium spatial (approx. 5–7 km) but high temporal (30 minute) resolution, gap-free data from MSG, with the low temporal (2–3 days) but high spatial (1 km) resolution of the Sentinel 3 LST retrievals. The resulting 1 km and daily LST dataset is based on an hourly merging of both datasets through bias-correction and Kalman Filter assimilation. Shortwave outgoing radiation is computed from the incoming shortwave radiation from MSG and downscaled albedo using 1 km PROBA-V data. MSG incoming shortwave and longwave radiation and the outgoing radiation components at 1 km spatial resolution are used together to compute the final daily SNR dataset in a consistent manner. Validation results indicate an improvement of the mean squared error by ca. 7% with an increase in spatial detail compared to the original MSG product. The resulting pan-European SNR dataset, as well as the merged LST product, can be used for hydrological modelling and as input to models dedicated to estimating evaporation and surface turbulent heat fluxes and will be regularly updated in the future. The datsets can be downloaded from https://doi.org/10.5281/zenodo.8332222 (Rains, 2023a) and https://doi.org/10.5281/zenodo.8332128 (Rains, 2023b).

*Copyright statement.* TEXT

# 1 Introduction

The Earth radiation budget describes how the Earth gains energy from the sun (shortwave radiation), and loses energy back to space through its reflection and the emission of thermal (longwave) radiation (Dewitte and Clerbaux, 2017; Kato et al., 2018). Due to the geometry of the Earth's orbit around the Sun, the yearly average net radiation at the bottom-of-atmosphere, namely the Surface Net Radiation (SNR), is positive at the equator and decreases towards the poles. This geographical energy imbalance is the main driver of the global atmospheric and oceanic circulation, which transports this energy surplus from the equator towards the poles (Dewitte and Clerbaux, 2017; Kato et al., 2018). SNR is thus a key driver in explaining the distribution of different climate regions and ecosystems on Earth (Köppen and Geiger, 1936), and it dominates the dynamics of biospheric and hydrological processes (Chapin et al., 2002). For this reason, SNR is used as forcing variable in many land surface models, hydrological models and satellite-based retrieval algorithms to estimate (e.g.) evaporation, runoff, soil moisture or surface heat fluxes.

The top-of-atmosphere radiation components can be derived directly from satellites. However, dynamic atmospheric (e.g., cloud and aerosol optical depth) and land (e.g. emissivity, LST, albedo or biomass) properties make it more challenging to obtain radiation estimates at the bottom-of-atmosphere, which are much more relevant to the above-mentioned biospheric and hydrological processes. As it is transmitted through the atmosphere, incoming shortwave radiation is scattered and absorbed by aerosols, gases and clouds, changing the temperature of the atmosphere and its emission of longwave radiation in all directions. The radiation reaching the surface is partly reflected depending on land cover and surface conditions and again interacts with the atmosphere/clouds once reflected. According to Stephens et al. (2012), on average 12% of the radiation reaching the surface is reflected back into the atmosphere; this is known as the surface planetary albedo. Then, part of the incoming radiation absorbed at the land surface is emitted towards the atmosphere as longwave radiation, as described by the Stefan–Boltzmann law. The modelling of these atmospheric and surface processes is required to obtain SNR – i.e. the balance between shortwave and longwave incoming and outgoing radiation at the surface – and it makes satellite-based SNR retrievals indirect and uncertain (Kato et al., 2018).

Over the past decades, numerous satellites/instruments have been launched to enable the monitoring of the radiation budget. Examples of programmes exploiting these observations to produce long-term global reliable estimates of the individual SNR components (i.e. shortwave and longwave, and both incoming and outgoing) are the International Satellite Cloud Climatology Project (ISCCP, Young et al. (2018)) and the Clouds and the Earth's Radiant Energy System (CERES) project (Wielicki et al., 1996). A comparison between the CERES product and radiation estimates from global reanalyses is given by Jia et al. (2018). Both satellite-based and reanalysis SNR products are mostly provided at a coarse (ca. $0.25°$) spatial resolution. This makes them suitable for global analysis or as input in global land surface models, but insufficient for most regional-scale studies. A few studies have already attempted to produce SNR data at higher spatial resolutions. For instance, Verma et al. (2016) proposed a method to yield a global 5 km SNR product at 8-day resolution by combining high-resolution variables derived from the Moderate Resolution Imaging Spectroradiometer (MODIS) Aqua satellite (including clear-sky LST, emissivity, aerosol optical depth and albedo) and a radiative transfer model with ancillary datasets from reanalysis. Also with a resolution of 5 km, Jiang

et al. (2016, 2018) developed the GLASS daily daytime net radiation product based on Multivariate Adaptive Regression Splines, combining incoming shortwave radiation, albedo and NDVI with further meteorological ancillary variables, such as wind speed, surface pressure and air temperature. Meanwhile Jiang et al. (2023) developed a methodology, based on Landsat data and ancillary datasets, using Machine Learning to produce daily net radiation at 30 m resolution. As an alternative to such methods, which are based on data from polar-orbiting satellites, to achieve a much higher temporal resolutions (sub-daily) at the expense of spatial resolution, observations from geostationary satellites can be used. The Satellite Applications Facility (LSAF) programme uses observations from the SEVIRI instrument onboard the Meteosat Second Generation (MSG) satellite to produce a SNR dataset at a spatial resolution of ca. 5–7 km (Trigo et al., 2011). These resolutions however appear still insufficient for regional water and agricultural management assessments in heterogeneous landscapes.

In this study, we present a 1 km SNR, and LST, dataset for Europe using MSG and polar orbiting observations. It is based on combining operationally available hourly incoming shortwave/longwave radiation retrievals from the above-mentioned LSAF programme at moderate (5–7 km) spatial resolution with hourly LSAF LST estimates as well as higher resolution (1 km) albedo retrievals from PROBA-V and LST from Sentinel 3 (Donlon et al., 2012). The novelty of this study lies in systematically exploiting the advantages, and mitigating the disadvantages, in terms of spatial and temporal resolution of available observations, which are well validated, in a physical and consistent manner based on the surface energy balance, and assembling a net radiation dataset from the individual incoming and outgoing radiation components. This includes the development of a 1 km gap-free LST product for downscaling outgoing longwave radiation. All-sky estimates are particular important for LST as cloud cover severely restricts the availability of clear-sky retrievals and it is temporally highly variable. This is underpinned by a number of previous studies which have focused on producing all-sky LST estimates, see e.g. Xu and Cheng (2021) and Jia et al. (2023), the latter also exploiting observations from geostationary and polar-orbiting products. 1 km albedo, for the computation of outgoing shortwave radiation, is equally calculated by combining polar and geostationary observations. The merged hourly SNR and LST data is for robustness resampled to daily time steps. The coarse-scale (5–7 km) all-sky LST estimates provided through the LSAF programme have only recently been released and the methodology here aims at exploiting these new data in an optimal manner. To our understanding, a systematic combination of these polar and geostationary retrievals with the overall goal of calculating a consistent high-resolution SNR product has not yet been undertaken. We argue that this approach based on the surface energy balance is the most consistent and, in theory, should yield the most accurate results.

The here presented published data is especially meant as a high-resolution forcing dataset for models which require SNR, such as The Global Land Evaporation Amsterdam Model (GLEAM). Such models can also benefit from high-resolution all-sky LST data making the intermediate merged LST product equally useful. In principle, the methodology can be extended to regions where the same variables are available from other geostationary and polar-orbiting satellites. The data and method are presented in detail in sections 2 and 3. All input and derived radiation components are validated against *in situ* measurements sites located across the study domain (section 4) and the SNR dataset is compared to ERA5-Land (Muñoz-Sabater et al., 2021). Finally, a discussion, in respect to similar studies, and concluding remarks are given in sections 5 and 6. The daily SNR and LST datasets are available for scientific use under https://doi.org/10.5281/zenodo.8332222 / https://doi.org/10.5281/zenodo.8332128 as netcdf

files (RNETdaily_lon_lat.nc and LSTdaily_lon_lat.nc), see Rains (2023a) and Rains (2023b). The spatial domain covered is -11.5 to 26.5 longitude and 35 to 71 latitude. The initial dataset is available for the years 2018–2019.

## 2 Data

Table 1 provides a general overview of the satellite data products used in this study. Shortwave and longwave incoming radiation components, $SW_{in}$ and $LW_{in}$, as well as emissivity $\varepsilon$, albedo $\alpha$ and LST are provided by LSAF (lsa-saf.eumetsat.int) and are based on observations from the Spinning Enhanced Visible and InfraRed Imager (SEVIRI) instrument onboard the Meteosat Second Generation (MSG) geostationary satellite. These MSG products are provided with a 30-minute sampling, but to reduce data volumes we base our methodology on hourly data. The spatial resolution across the European domain is approximately 5–7 km depending on latitude. In addition, 1 km LST retrievals from the Sea and Land Surface Temperature Radiometer (SLSTR) instrument onboard Sentinel 3 as well as 1 km albedo retrievals from PROBA-V are used to compute the high-resolution LST dataset and outgoing radiation components. For the purpose of validation, we use radiation measurements from sites distributed across Europe belonging to different international networks. A more detailed description of the satellite retrievals and *in situ* data used in the study is provided in the following subsections. Note as well that ERA5-Land (Muñoz-Sabater et al., 2021) is also used in section 4 for comparison purposes.

| Variable | Satellite | Orbit | Temporal | Spatial | Coverage |
|---|---|---|---|---|---|
| $SW_{in}$ | MSG | geostationary | hourly | 5–7 km | all-sky, clear-sky+model |
| $LW_{in}$ | MSG | geostationary | hourly | 5–7 km | all-sky, clear-sky+model |
| LST | MSG | geostationary | hourly | 5-7 km | all-sky, clear-sky+model |
| LST | Sentinel 3A | polar | 2–3 days | 1 km | clear-sky |
| $\varepsilon$ | MSG | geostationary | daily | 5–7 km | clear-sky composite |
| $\alpha$ | MSG | geostationary | daily | 5–7 km | clear-sky composite |
| $\alpha$ | PROBA-V | polar | 10-daily | 1 km | clear-sky composite |

**Table 1.** Overview of satellite based products used in the study with their respective temporal and spatial resolution as well as their coverage, i.e. clear-sky vs. all-sky.

### 2.1 Incoming shortwave/longwave radiation

We use hourly data from the LSAF programme, part of the distributed Applications Ground Segment SAF network serving as the European organisation for the Exploitation of Meteorological Satellites (EUMETSAT). The data are based on observations provided by SEVIRI onboard MSG, acquired at 12 spectral channels with 3 km resolution at nadir (1 km for the high-resolution visible channel) (Trigo et al., 2011). A detailed description of the LSAF methodology on deriving $SW_{in}$ and its validation is

given by Carrer et al. (2019a) and Carrer et al. (2019b). Details on the estimation and evaluation of $LW_{in}$ are given by Trigo
et al. (2010) and Carrer et al. (2012).

## 2.2 LST

The LSAF all-sky LST product based on the SEVIRI instrument onboard the geostationary Meteosat Second Generation
(MSG, Martins et al. (2019)) is a combination of the clear-sky MSG level 2 product, MSLT (LSA-001), based on a Generalised
Split-Window (GSW) algorithm (Trigo et al., 2008a), and output from an energy balance algorithm which is also used for
the production of the MSG 30-minute evaporation (MET-v2, LSA-311) dataset (Ghilain, 2016). The energy balance algorithm
incorporates other LSAF SEVIRI-based products such as shortwave and longwave radiation fluxes, land surface albedo or veg-
etation, soil moisture based on the assimilation of scatterometer observations provided by the Hydrology SAF (H-SAF), and
near surface meteorological information obtained from the European Centre for Medium-Range Weather Forecasts (ECMWF)
operational forecasts (Ghilain et al., 2020). Within the model, each pixel is composed of different tiles representing a particular
surface type based on the ECOCLIMAP-II database (Faroux et al., 2013). Pixel values are computed from the weighted average
of the four most dominant tiles. The advantage of using geostationary satellites is the high temporal resolution, allowing for
the characterisation of the LST diurnal cycle. An assessment of the accuracy of the LST is given by Martins et al. (2019). The
product comes with gridded uncertainty estimates, which are used in the LST merging procedure.

Higher-resolution, clear-sky LST estimates are obtained from Sentinel 3. The Sentinel 3 mission consists of two polar-
orbiting satellites (Sentinel 3A/B) launched on February 16, 2016, and April 25, 2018 (Ghent et al., 2017; Zheng et al.,
2019; Nie et al., 2021), both carrying the Sea and Land Surface Temperature Radiometer (SLSTR) instrument. They have
a revisit time of 2–3 days. The instrument has nine channels, three of them covering the visible and near-infrared (VNIR)
part of the spectrum, three the shortwave infrared (SWIR), and the remaining three the middle-infrared (MIR and TIR, Nie
et al. (2021)). For this study, we use the Climate Change Initiative (CCI) LST product provided at a spatial resolution of 0.01
degrees (https://climate.esa.int/en/odp//project/land-surface-temperature). Included in the product is the exact overpass time
and as for the LSAF LST from MSG the total estimated uncertainty for each retrieval, necessary for the merging of the polar
and geostationary LST data. For this initial study focusing on 2018–2019 only Sentinel 3A data was used. Sentinel 3B was
launched in April 2018 and flown in tandem with Sentinel 3A from June to October of the same year after which it was moved
to its nominal orbit (Clerc et al., 2020). The approximate local overpass time of Sentinel 3A and Sentinel 3B thereafter is the
same (ca. 10:30 am/pm) with the precise time varying and taken into account in the merging methodology (see section 3.3).

## 2.3 Surface emissivity

Land surface $\varepsilon$ is required, in conjunction with LST, to calculate $LW_{out}$. Approaches to retrieve $\varepsilon$ can be broadly separated into
methods where LST and $\varepsilon$ are jointly retrieved or where $\varepsilon$ is retrieved in isolation. The latter was initially used within the LSAF
programme, and relied on spectral data for the various land covers based on spectral libraries, and dynamic land cover fractions
(Peres and DaCamara, 2005). To overcome difficulties linked to performing the retrieval of LST and $\varepsilon$ separately under certain

conditions, e.g. in semiarid regions, LST and $\varepsilon$ are now simultaneously retrieved by the LSAF programme including for the products we use in this study (Trigo et al., 2008b).

## 2.4 Albedo

The LSAF $\alpha$ product based on the MSG SEVIRI instrument is produced following three steps: (1) an atmospheric correction of top-of-atmosphere measurements to obtain reflectances, (2) a daily inversion of a semi-empirical model of the bidirectional reflectance distribution function, and then the consideration of all inversions within a temporal window to reduce the impact of outliers and reduce data gaps, and (3) the angular integration for each channel and the spectral integration (Geiger et al., 2008; Carrer et al., 2018). The product thus describes the hemispherical broadband $\alpha$. As a second hemispherical broadband $\alpha$ product, we use 1 km retrievals based on ProbaV and distributed through the Copernicus Global Land Service (CGLS). The retrieval follows the same methodology as for the LSAF $\alpha$ product.

## 2.5 *In situ* measurements

For the validation of the merged daily SNR dataset and the individual radiation components we use radiation measurements taken at a total of 73 sites distributed across Europe for the 2-year study period (2018–2019). Measurements are obtained from the Baseline Surface Radiation Network (BSRN) (Driemel et al., 2018), the European Fluxes Database Cluster (http://www.europe-fluxdata.eu, EFDC), the Integrated Carbon Observation System (ICOS) (Heiskanen et al., 2021), the FLUXNET-CH4 network (Delwiche et al., 2021), and SAPFLUX (Poyatos et al., 2021). Table A, see appendix A, provides a comprehensive list of the in-sites used for this study. For a number of sites all radiation components are available (54) while for others only a subset is available. The Table includes the station ID, name, geographic coordinates and IGBP land cover class as well as which radiation components are available for validation. The following land cover classes are covered: Cropland (CRO), closed shrublands (CSH), deciduous broadleaf forest (DBF), evergreen needleleaf forest (ENF), grassland (GRA), mixed forest (MF), open shrublands (OSH), savanna (SAV), urban (URB), wetland (WET) and woody savanna (WSA).

While the *in situ* measurements are considered as ground-truth, it is necessary to mention that they have their own sources of uncertainties. Incoming shortwave and longwave radiation are measured by pyranometers and pyrgeometers. Accuracy targets for the BSRN network measurements (from 2004) are for example 2% or 5 W m$^{-2}$ for incoming shortwave radiation and 2% or 3 W m$^{-2}$ for incoming longwave radiation. Target uncertainties for outgoing shortwave and longwave radiation are 3% and 2% (or 3 W m$^{-2}$) respectively (McArthur, 2004). For the measurement of the outgoing radiation components the pyranometer/pyrgeometer is installed facing downwards. The target uncertainties are in line with the achievable accuracy of the pyranometer/pyrgeometer instruments although they might not be met under some conditions, e.g. incorrect installation at an angle or snow cover. The instruments should be calibrated, e.g. every 2 years (Walter-Shea et al., 2019).

## 3   Methodology

### 3.1   SNR calculation

SNR is computed using the radiation balance equation (1).

$$SNR = (SW_{in} + LW_{in}) - (SW_{out} + LW_{out}) \tag{1}$$

where $SW_{in}$ is hourly incoming shortwave radiation (W m$^{-2}$) and $LW_{in}$ is hourly incoming longwave radiation (W m$^{-2}$), both from LSAF (see section 2). $SW_{out}$ and $LW_{out}$ are hourly outgoing shortwave and outgoing longwave radiation (W m$^{-2}$), respectively, calculated as:

$$SW_{out} = SW_{in} * \alpha \tag{2}$$

$$LW_{out} = \varepsilon * \sigma * LST^4 + (1 - \varepsilon) * LW_{in} \tag{3}$$

with $\sigma$ being the Stefan–Boltzmann constant (i.e. 5.67 x $10^{-8}$ W m$^{-2}$ K$^{-4}$). Both $SW_{out}$ and $LW_{out}$ are to a large degree controlled by land surface properties and processes, i.e. $SW_{out}$ by $\alpha$ (equation 2), and $LW_{out}$ by $\varepsilon$ and LST (equation 3). LST, in particular, dictates the magnitude and variability of $LW_{out}$ over different spatial and temporal scales. Note that the term $(1 - \varepsilon) * LW_{in}$ accounts for longwave reflection (Maes and Steppe, 2012).

The focus here is on the improvement of the spatial resolution of the LSAF $SW_{out}$ and $LW_{out}$ by using gap-free all-sky 1 km $\alpha$ and $LST$ in equations 2 and 3, respectively. The details of these datasets are given in section 3.2 and 3.3. The rationale is based on the assumption that $SW_{out}$ and $LW_{out}$, especially on the daily scale which we aggregate to, are spatially more heterogeneous than the incoming components. Therefore, by using higher-resolution $\alpha$ and $LST$, the final SNR dataset can better capture the variability induced by landscape features and conditions.

### 3.2   Bias correction of albedo

To obtain a spatially and temporally gap-free $\alpha$ dataset at 1 km resolution, we bias-correct the daily $\alpha$ from LSAF towards the retrievals from ProbaV using the mean of the temporally overlapping retrievals for 2018–2019. Remaining gaps are filled through linearly interpolating/extrapolating based on the nearest data points in the temporal domain. Prior to the bias correction, the $\alpha$ products are regridded using nearest-neighbour interpolation to a common 0.01° grid. Since both sets of $\alpha$ are based on the same methodology, we assume that the bias can be largely attributed to the difference in spatial resolution, but also the MSG product integrating multiple observations per day, and possibly to the differences in the channels (ProbaV and SEVIRI response functions).

## 3.3 Merging of LST

The merging of the hourly LSAF LST (5–7 km) and Sentinel 3 LST (1 km) relies on the assumption that the diurnal cycle of LSAF is reliable in relative terms, whereas the Sentinel 3 LST can be trusted in absolute terms. This approach allows us to benefit from the high temporal resolution of the geostationary data and the high spatial resolution of the Sentinel 3 observations. The all-sky LSAF product, which contains modelled LST when cloud cover prevents the direct retrieval, enables the merged gap-free LST product with Sentinel-3 resolution. After regridding the LSAF observations, using nearest-neighbour interpolation, to the 0.01° grid of Sentinel 3 observations, we follow a stepwise approach:

1. Temporal normalisation of Sentinel 3 daytime/nighttime observations on the hour.

   The Sentinel 3 LST is available every ~2–3 days both during daytime (~10 am local time) and nighttime (~10 pm local time), conditioned on the presence of clear-skies. However, because of slightly differing overpass times from day to day we first normalise the Sentinel 3 daytime/nighttime observations individually to on the hour (e.g. 10:00 for daytime), using information from the diurnal cycle described by the hourly LSAF observations of the same day. For that, at each grid cell, we convert the on the hour daytime and nighttime overpass time of the Sentinel 3 observations from local time to UTC. Then, when a Sentinel 3 daytime or nighttime observation is acquired, e.g. prior to that mean UTC daytime or nighttime overpass hour $t$, the observation is corrected through linear interpolation using the LSAF LST retrievals at $t$ and the previous hour $t-1$ on that day:

   $Sentinel3LST_{nor} = Sentinel3LST + \Delta t * (LSAFLST_t - LSAFLST_{t-1})$

   with $\Delta t$ being the difference between the on the hour mean nighttime/daytime overpass time $t$ and the exact overpass time of the specific Sentinel 3 observation on that day. We do not perform the linear interpolation if $LSAFLST_{t-1}$ and/or $LSAFLST_t$ are not clear-sky observations, i.e. the pixel is covered by cloud, and in that case, we disregard the Sentinel 3 observation. This is based on the assumption that the diurnal cycle will be less accurate when mixing clear-sky/all-sky estimates or only relying on modelled all-sky estimates. Sentinel 3 observations with a $\Delta t$ of more than 45 minutes (i.e. $\Delta t$>0.75) are equally excluded to reduce errors from the linear interpolation.

2. Bias-correction of daytime/nighttime LSAF observations towards the normalised, high spatial resolution, Sentinel 3 daytime/nighttime observations.

   The previously individually normalised Sentinel 3 observations $Sentinel3LST_{nor}$ are used as the basis to bias-correct the geostationary observations at the same mean on the hour overpass time $t$ (daytime and nighttime separately) per grid cell using the means based on overlapping $Sentinel3LST_{nor}$ and $LSAFLST_t$ observations for the entire 2018–2019 record.

3. Bias-correction of the entire hourly geostationary $LSAFLST$ time series per grid cell by assuming that the bias corrected for in the previous steps applies to the subsequent hourly observations too.

   We apply the bias that was applied to the geostationary daytime observations at the mean Sentinel 3 overpass time to all

hours of the same day after the mean Sentinel 3 overpass time and until the mean Sentinel 3 nighttime overpass time. We apply the nighttime bias correction for the hourly observations until next daytime overpass time.

4. Assimilation of the normalised Sentinel 3 observations $Sentinel3LST_{nor}$ from Step 1 into the bias-corrected hourly geostationary LSAF LST time series from Step 3.

   At a given pixel and point in time when both $LSAFLST$ and $Sentinel3LST_{nor}$ are available, the bias-corrected geostationary LST ($LSAFLST$) is updated. This is done taking into account the uncertainty of both sets of observations using a Kalman Filter:

   $$LSAFLST_a = LSAFLST + K(LSAFLST - Sentinel3LST_{nor})$$

   where $LSAFLST_a$ is the updated LST at the hour $t$ and $K$ is the Kalman gain with the range [0, 1], computed as:

   $$K = PH^T(HPH^T + R) - 1$$

   with $P$ being the uncertainty of the geostationary observation $LSAFLST$ and $R$ the uncertainty of the Sentinel 3 observation at time step $t$. Both uncertainties are available for each individual pixel and time-step. $H$, the observation operator, is 1 as there is no difference between model and observation space. Normally, the update in a Kalman Filter is propagated over time through a dynamic model. Here, there is no such prognostic model to predict LST, thus we correct all subsequent hourly $LSAFLST$ observations by the same amount until the next Sentinel 3 observation is available. Some more details about the LST merging and the Kalman filtering step are given in appendix F.

## 4 Analysis and validation

### 4.1 Incoming radiation fluxes

Comprehensive validation studies in literature against pyranometer measurements show the high accuracy of the LSAF radiation products; see e.g. Carrer et al. (2019b) or Lopes et al. (2022). A validation of the LSAF $SW_{in}$ data by Roerink et al. (2012) against the CarboEurope flux tower network shows a very high accuracy, corroborated by comparing the satellite product with available radiation estimates from about 300 operational weather stations. Our own validation of both the LSAF $SW_{in}$ and $LW_{in}$ products shows a similar good performance, with Pearson's correlation coefficients consistently above 0.9. Figure 1 (top panels) show the correlation coefficients for all *in situ* sites in Europe for the 2018–2019 period. They are generally higher for $SW_{in}$ than for $LW_{in}$. In terms of the root-mean-squared error (RMSE), $SW_{in}$ and $LW_{in}$ perform similarly across all sites. Few stations with a considerably worse match between observations and *in situ* data are located in Belgium for $SW_{in}$, and around the Alps for $LW_{in}$. It is fair to consider that the temporal variability of cloud cover determines to a large extent the variability of $SW_{in}$ and $LW_{in}$. This is also the main information provided by satellite data (clouds and cloud optical depth via top-of-atmosphere reflectances). So the generally high R values for both $SW_{in}$ and $LW_{in}$ corroborate that satellite products follow reasonably well the in situ time-series. $LW_{in}$ estimates require screen variables ($LW_{in}$ is more indirectly linked with

top-of-atmosphere observations than $SW_{in}$), which are derived from Numerical Weather Prediction models. Therefore it is not surprising that R and RMSE are not as good as those for $SW_{in}$. The accuracy of screen variables may also explain the worse performances of $LW_{in}$ in the Alps due to the very high spatial heterogeneity. Although some orographic corrections are performed, the uncertainty is generally likely larger in mountainous regions. Since the availability of *in-situ* measurements is already fairly limited, we argue that carrying out the validation also in challenging terrain benefits the overall accuracy assessment. Figure 2 shows both $SW_{in}$ and $LW_{in}$ for two example sites, namely BE-Dor and IT-Lsn.

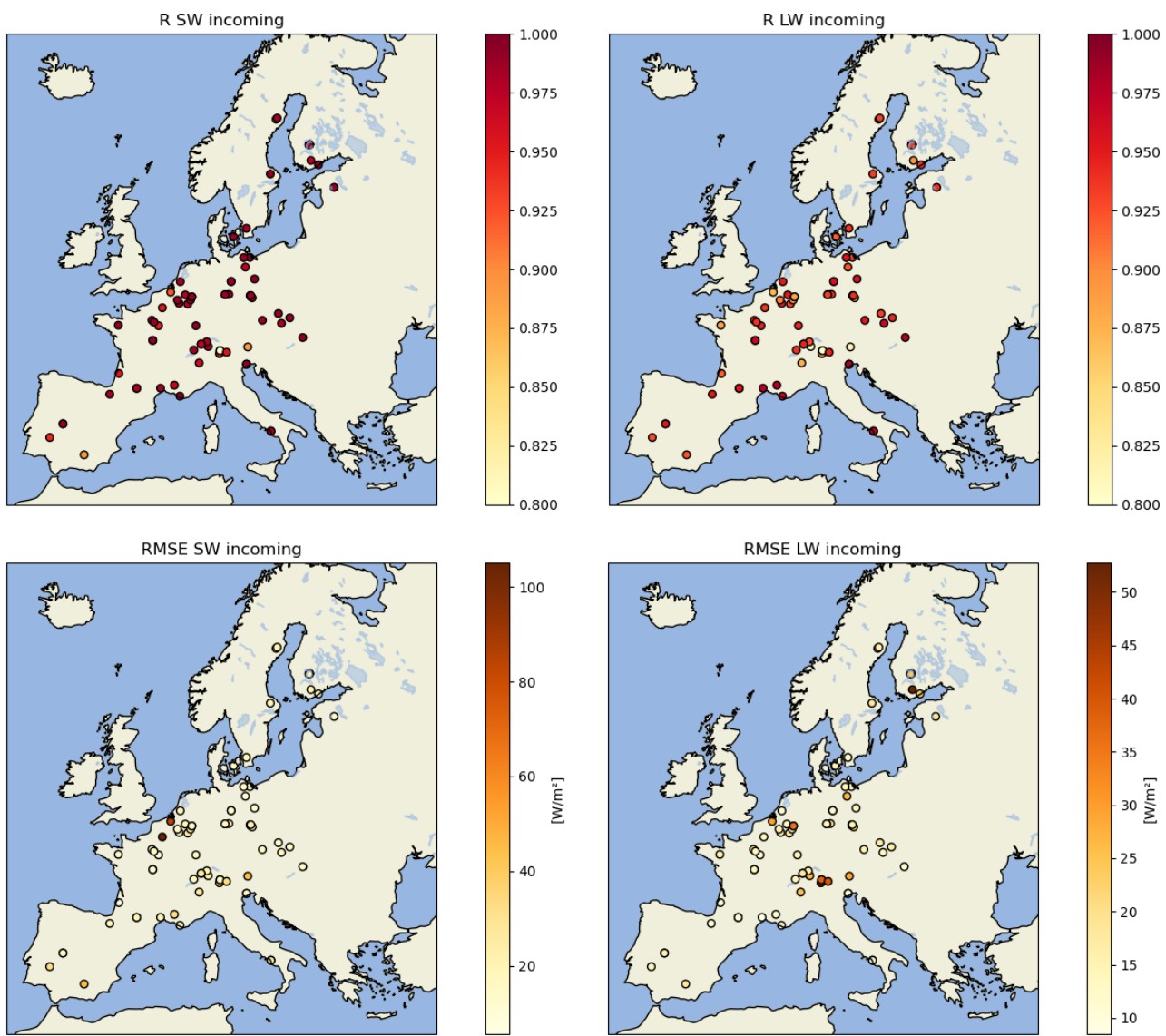

**Figure 1.** Validation of $SW_{in}$ and $LW_{in}$ from LSAF across Europe for 2018–2019 in terms of Pearson's correlation coefficient (R, top panels) and root mean squared error (RMSE, lower panels).

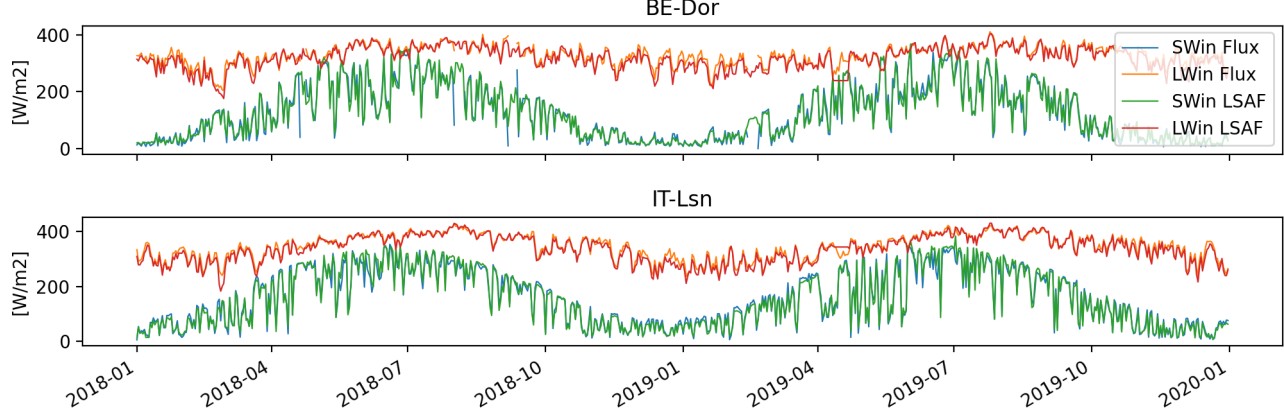

**Figure 2.** Daily averages of $SW_{in}$ and $LW_{in}$ from LSAF and ground truth for two stations BE-Dor and IT-LSN.

Additional seasonal validation statistics for the incoming radiation components are given in the appendix (see boxplots in Figures B1 and B3). In summary, for $SW_{in}$, R is consistently high throughout the year albeit with a higher spread of values for the individual seasons (given that the overall seasonal amplitude has a lesser impact). The Root-Mean-Squared-Error (RMSE) varies slightly from season to season with the highest values in summer (April/May/June and July/August/September). This coincides with generally much higher radiation values during these months. In terms of Mean-Square-Percentage-Error (MSPE) the error is highest in the winter months. A slight bias of 5 $W/m^2$ is observed throughout the year although it is less pronounced during winter and spring. Validation metrics for different land cover types are also given (Figures B2 and B4) with the ESA CCI land cover product (Defourny et al., 2023) being used as its spatial resolution (300m) is more consistent with the spatial resolution of the here developed data products than the land cover information provided by the FLUXNET sites. For $LW_{in}$ (Figures B2 and B4), R again shows a higher spread for the individual seasons than for the entire study period. RMSE is highest in spring. In terms of land cover, all land cover types show high values for R whereas For RMSE, RMSPE and bias the flooded/brakish/water areas clearly show degraded performance (B4).

## 4.2 Land surface temperature

Extensive validation of the LSAF and Sentinel 3 LST products has already been performed. Both have an average accuracy below 1.5 K, although it varies across space and time. Our goal is to combine their individual strengths in terms of spatial and temporal resolution to obtain an enhanced representation of landscape heterogeneity. For an in-depth quantitative validation of the Sentinel 3 LST product we refer to Pérez-Planells et al. (2021). The LSAF LST products were validated by Trigo et al. (2008a), Göttsche et al. (2013), Göttsche et al. (2016), Martins et al. (2019) and Trigo et al. (2021). Here the validation against *in situ* data is carried out not directly on LST but on $LW_{out}$ – see section 3.3. This is based on LST validation data being limited and a validation using $LW_{out}$ ground truth measurements thus being much more comprehensive. Furthermore, the developed LST product primarily serves the purpose of enabling a spatially downscaled $LW_{out}$ product for the final calculation of SNR.

Figure 3 shows a comparison between the mean annual LST for 2018–2019 from LSAF and the merged LSAF/Sentinel 3 LST for two regions in Europe. The downscaled LST product shows significantly more spatial detail, especially in heterogeneous or topographic complex areas such as the Central System in Madrid (top row) or the Rhine Valley and its surrounding mountainous areas (bottom row). Instead of the 2018–2019 LST average, Figure 4 shows the original LSAF LST and the downscaled LST product for 30th June 2018. This day was chosen for no particular reason and is representative for other dates.

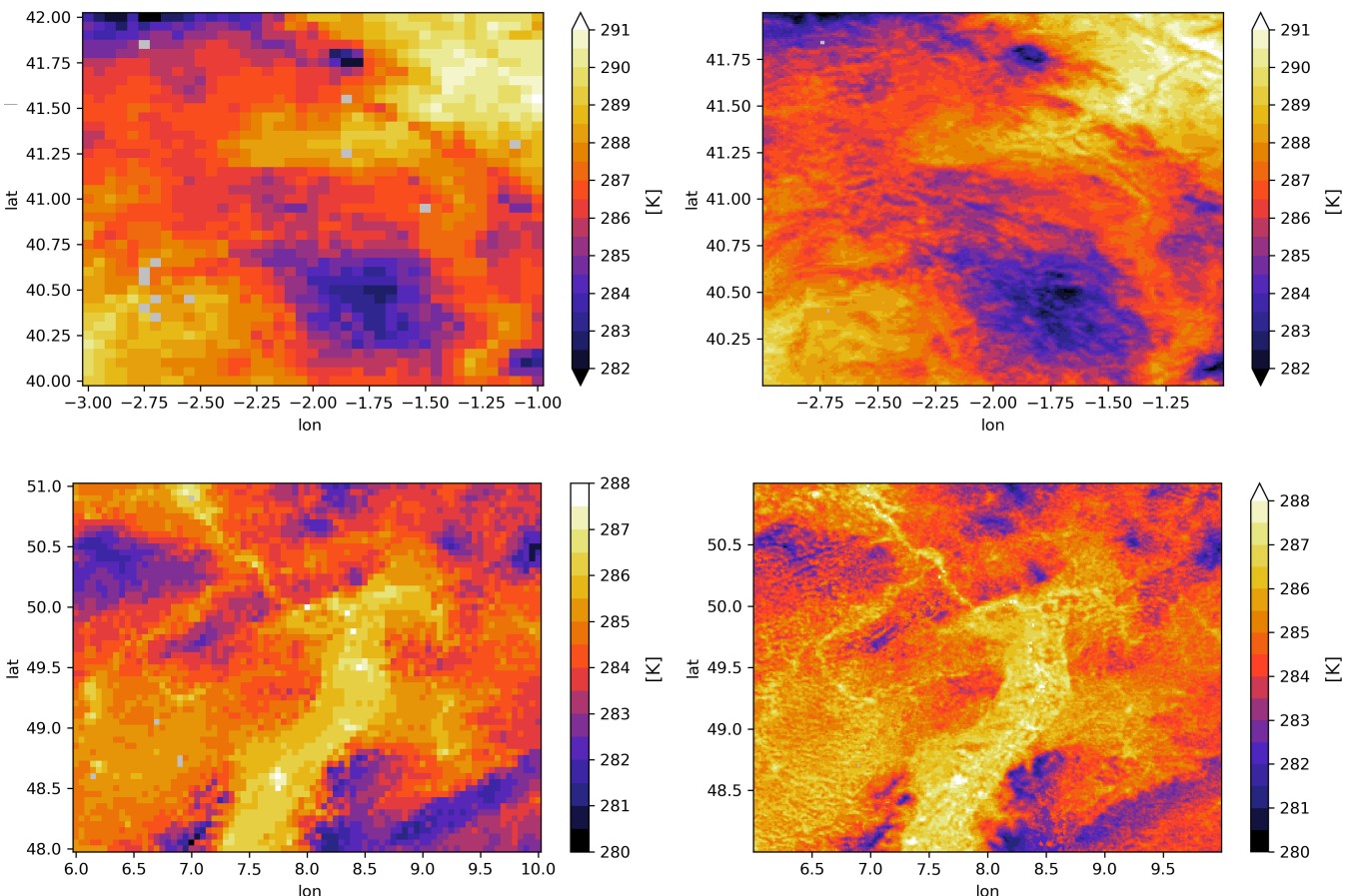

**Figure 3.** Mean LSAF LST (left) and merged LSAF/Sentinel 3 LST (right) for 2018–2019, showing a part of the Iberian Peninsula (top) and the southern Rhine Valley (bottom).

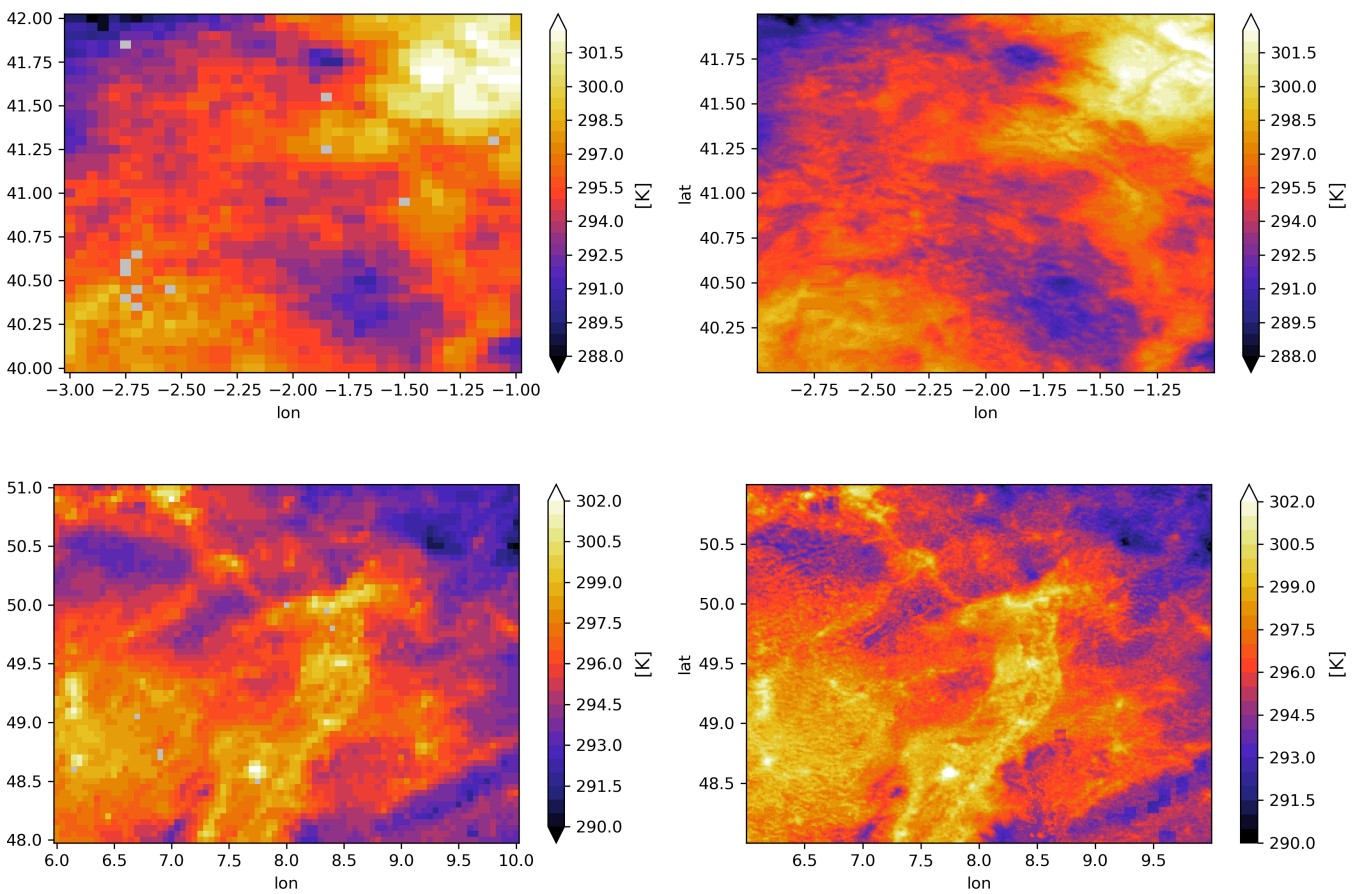

**Figure 4.** LSAF LST (left) and merged LSAF/Sentinel 3 LST (right) for 30th June 2018, showing the centre of the Iberian Peninsula (top) and the southern Rhine Valley (bottom).

## 4.3 Land surface albedo

Figure 5 shows the 2018–2019 mean albedo from LSAF and from the downscaled albedo product across parts of the Rhine valley, as well as the values for a single day, analogous to the LST figures 3–4. The effect of the downscaling in enhancing the spatial detail of the LSAF albedo retrievals based on PROBA-V retrievals is evident; see (e.g.) the distinct areas of low albedo surrounding the Rhine valley covered by forests and the higher albedo areas within the valley.

300

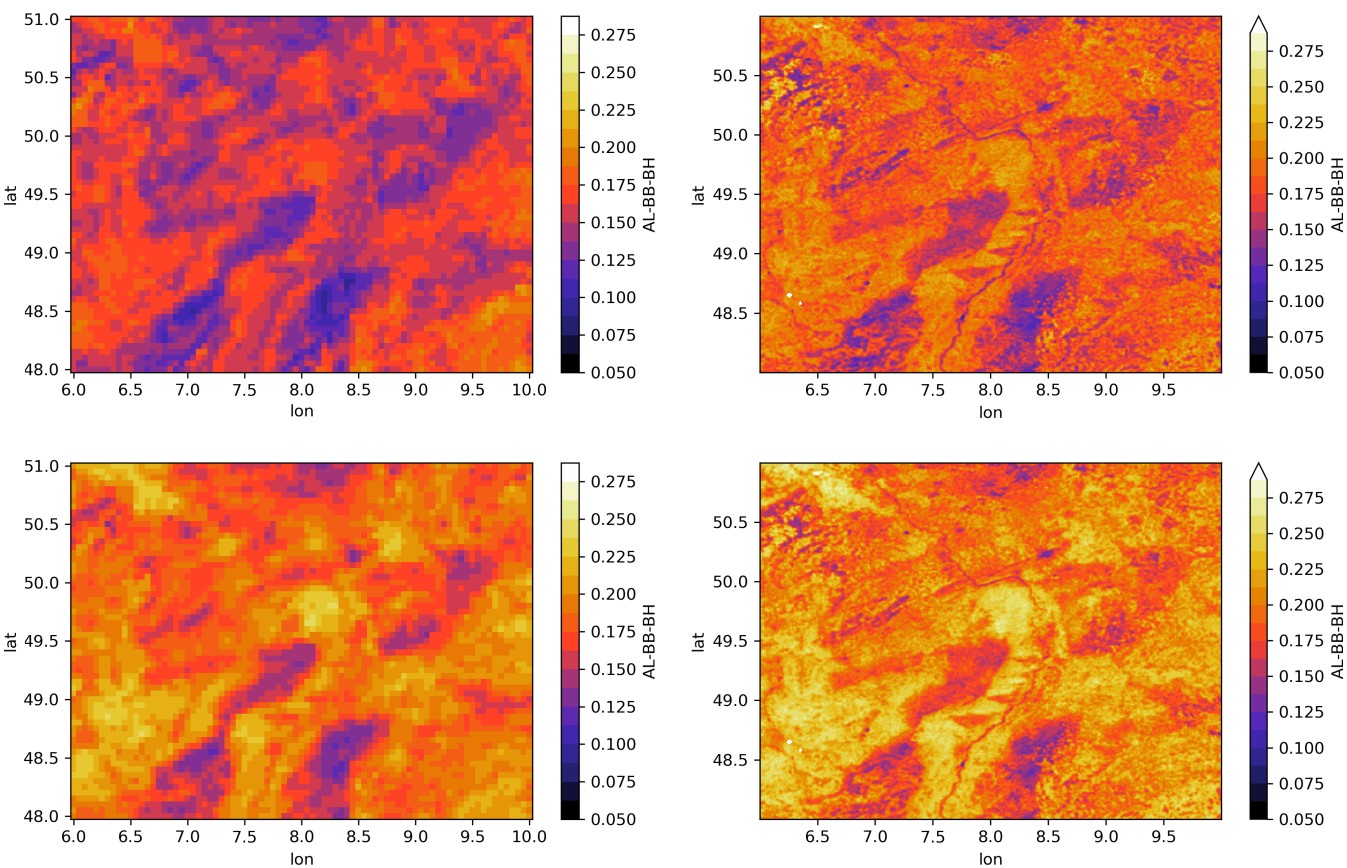

**Figure 5.** Mean albedo from LSAF (top left) and the downscaled dataset (top right) for 2018–2019, as well as the retrievals for the 30th June 2018 for LSAF (bottom left) and the downscaled albedo product (bottom right). The maps depict the southern Rhine valley with the river flowing from South to North through the centre of the landscape shown and then to the North-West.

### 4.4 Outgoing radiation fluxes

$SW_{out}$ estimates, resulting from combining LSAF $SW_{in}$ with either LSAF $\alpha$ or with the downscaled $\alpha$ dataset, are validated against *in situ* data. Likewise, $LW_{out}$, using either LSAF LST or the downscaled LST product, are also compared against *in situ* data. This validation therefore shows to what extent the downscaling of $SW_{out}$ and $LW_{out}$ in combination with emissivity data from LSAF influences the accuracy, and not only spatial detail, as shown in sections 4.2 and 4.3.

On average, both RMSE for $SW_{out}$ and $LW_{out}$ are lower when compared to using data from LSAF only, with a mean of 17.1 $W/m^2$ vs. 17.8 $W/m^2$ for $SW_{out}$, and 11.4 $W/m^2$ vs 11.04 $W/m^2$ for $LW_{out}$). Figure 6 shows the distribution of the RMSE across the available sites for the 2018–2019 period for $SW_{out}$ and $LW_{out}$. The absolute values for the RMSE of LSAF as well as the difference to the downscaled products are included.

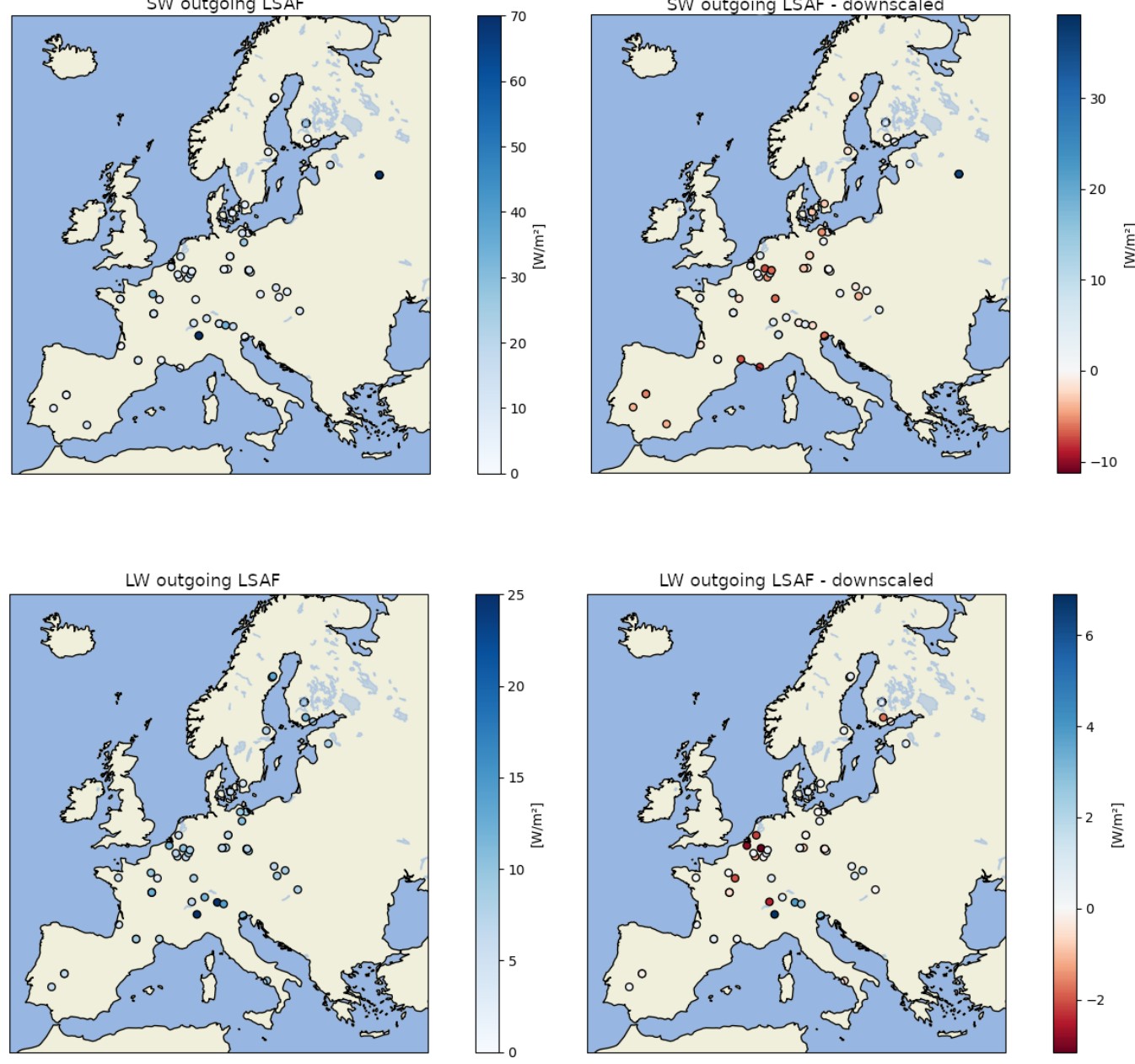

**Figure 6.** Validation of $SW_{out}$ (top) and $LW_{out}$ (bottom) in terms of RMSE. Based on LSAF only (left) and the difference to the downscaled products on the right; blue colours on the right panels indicate a better performance of the downscaled products.

Figure 7 shows R, MSE, MSPE and bias for LSAF and the downscaled product across the different CCI land cover types. For R, both $SW_{out}$ and $LW_{out}$ show a lower performance for the water related land cover types (see also incoming radiation validation). For MSE the same is true only for $SW_{out}$ and here tree covered areas show a slight positive bias whereas the over

land cover types are on average negatively biased. For $LW_{out}$ the bias seems less pronounced and the land cover median values are generally above or close to 0.

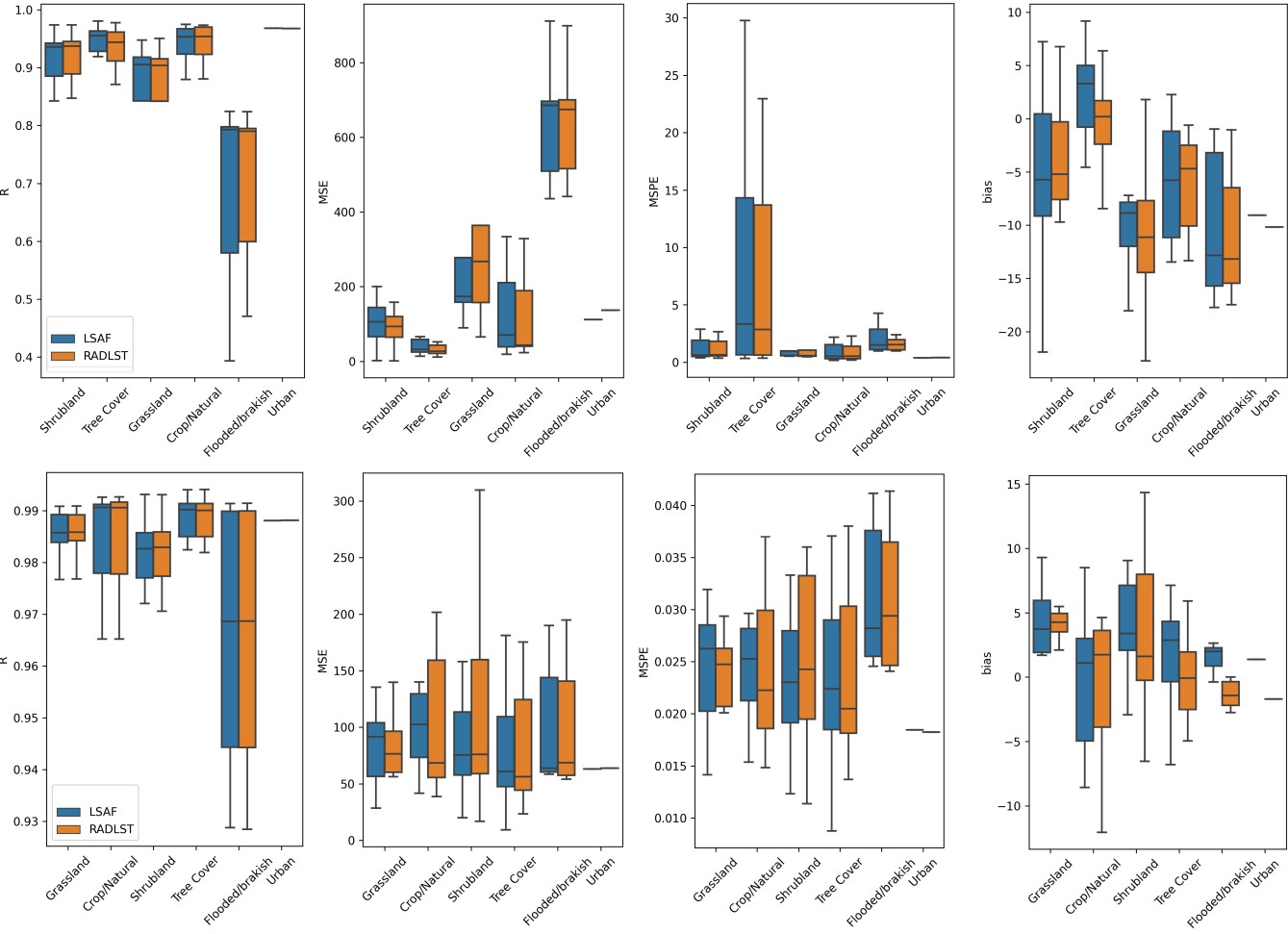

**Figure 7.** Validation of $SW_{out}$ (top) and $LW_{out}$ (bottom) radiation in terms of R, RMSE, RMSPE and bias for LSAF only and the downscaled product across different land cover types.

For a complete picture, the validation metrics are also calculated seasonally (see Figure C1 in annex). Seasonal patterns are most pronounced for RMSPE for $SW_{out}$, which is significantly higher during the winter months. One explanation is that the calculation relies on accurate albedo values but their retrieval is especially challenging in winter due to cloud cover. Valid albedo values are linearly interpolated to fill in the data gaps and especially snow cover will have a significant impact. High errors for $SW_{out}$ in snow cover conditions can thus be expected.

## 4.5 Surface net radiation

Finally, the downscaled SNR dataset, resulting from the hourly $SW_{in}$ and $LW_{in}$ as well as the downscaled hourly $SW_{out}$ and $LW_{out}$, is validated against the available *in situ* data at daily time scales. On average, the downscaled product has a RMSE of 22.53 $W/m^2$ vs 23.5 $W/m^2$ for the MSG only product. Figure 9 shows the distribution of RMSE values across the study domain. A time series for a single example site is shown in Figure 10. We also analyse how the downscaled SNR product performs under cloudy and clear-sky conditions. Clear-sky conditions were assumed for the daily SNR product when more than 12 hours of LSAF clear-sky LST observations were available. Figure 8 shows that for clear-sky conditions both R and the bias are improved when compared to cloudy conditions. The RMSE is slightly higher for clear-sky conditions, likely linked to seasonality as clear-sky conditions are more common during summer where also the SNR values are higher.

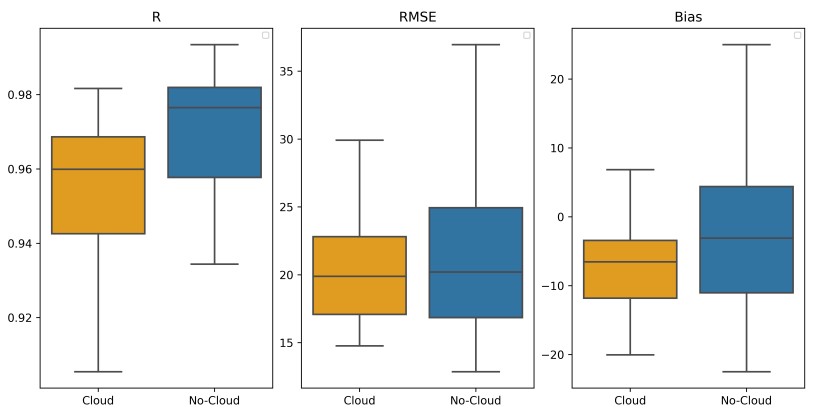

**Figure 8.** Validation of SNR for cloudy and clear-sky days in terms of R, RMSE and bias.

Figure 11 shows the SNR validation for the different CCI land cover types for a LSAF only based SNR as well as the downscaled product. The Figure also includes performance metrics for the ERA5-Land product (Muñoz-Sabater et al., 2021) which were included to give some context. R is generally high for all products (ca. 0.95) for all sites with the exception of sites with land cover affected by water. There ERA5-Land outperforms the LSAF and downscaled SNR product in terms of R, likely due to a sub-optimal treatment of these areas in the processing of the input products. In terms of MSE ERA5-Land again outperforms the other products for water affected land cover. However, for the other land cover classes the LSAF SNR and downscaled products perform better with the downscaled dataset showing the lowest values. In terms of bias, ERA5-Land performs best with the downscaled data performing between ERA5-Land and the LSAF only SNR.

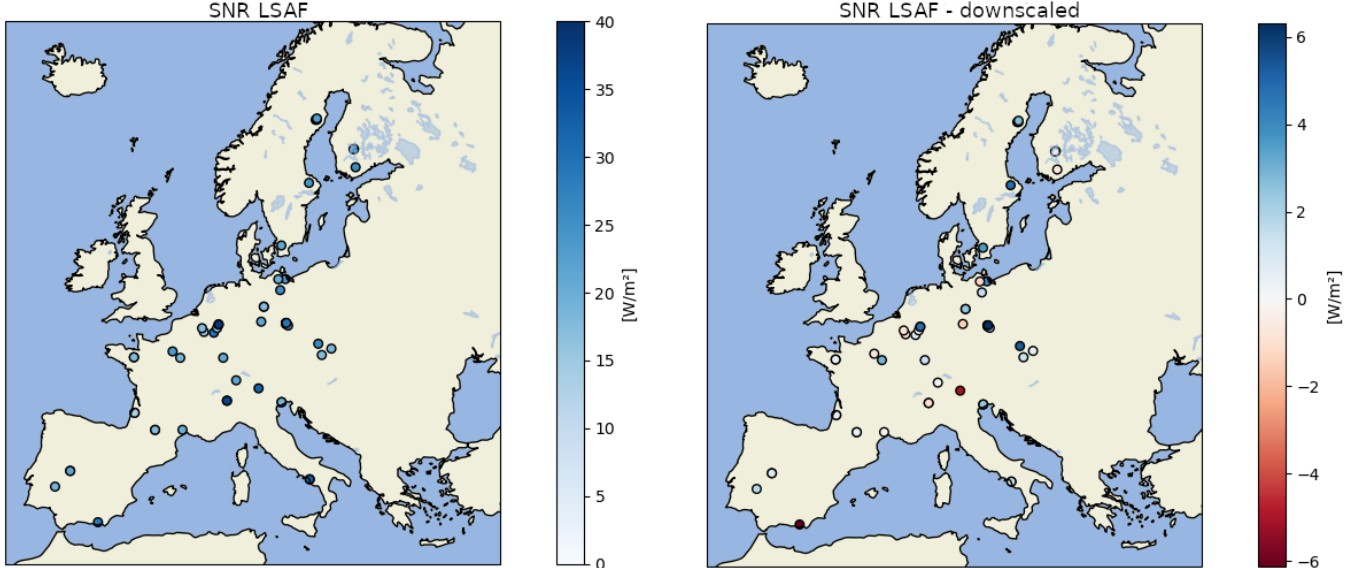

**Figure 9.** Validation of SNR in terms of RMSE using LSAF only (left) and the difference to the downscaled product on the right; blue colors on the right map indicate a better performance of the downscaled product.

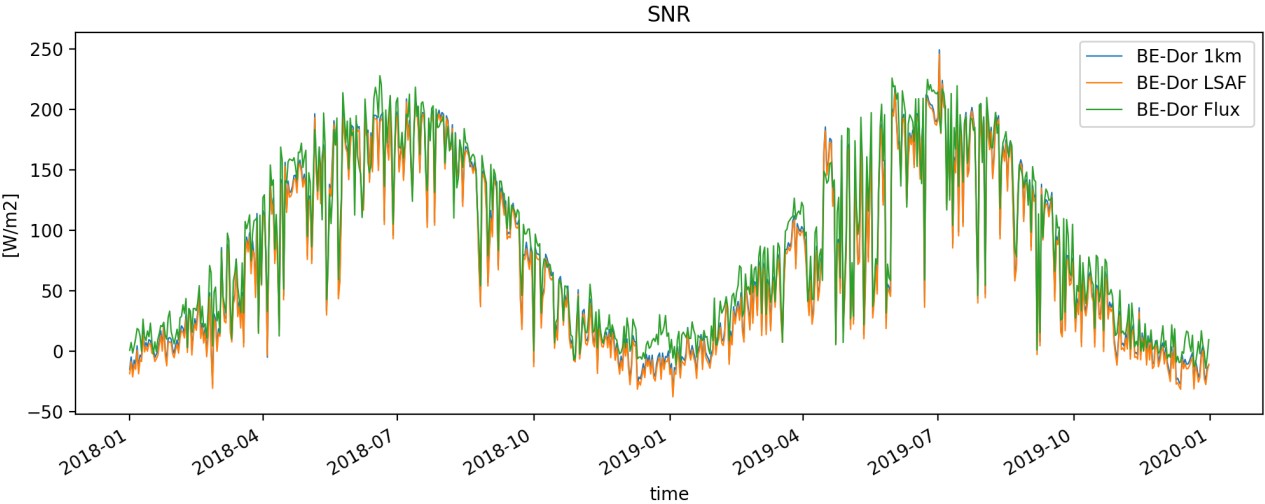

**Figure 10.** Daily averages of downscaled, LSAF SNR and ground truth for site IT-Lsn.

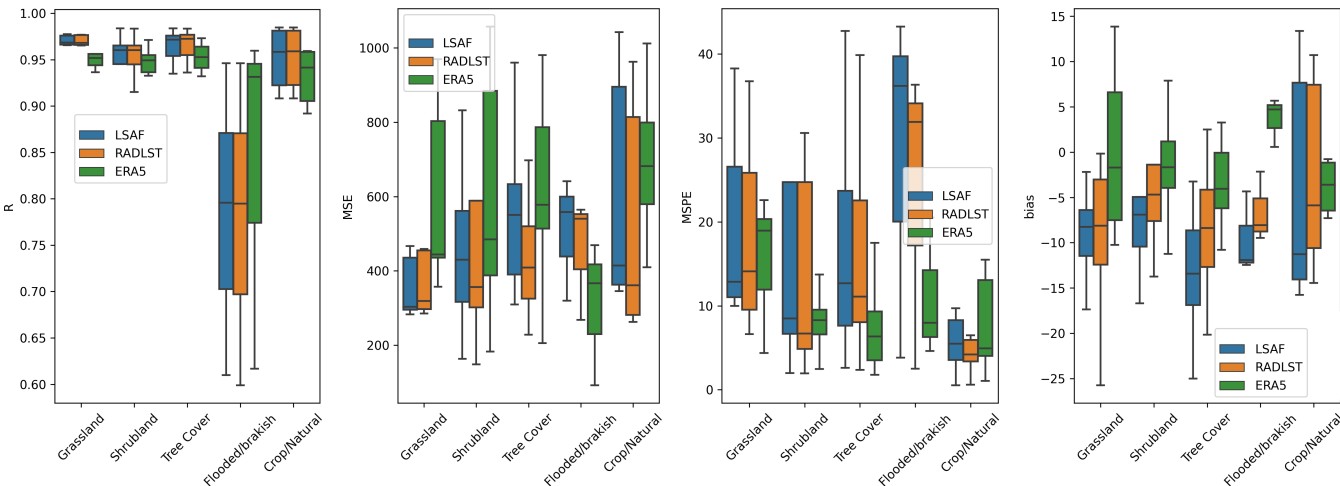

**Figure 11.** Validation of SNR for different CCI land cover types in terms of R, MSE, MSPE and bias.

For the SNR products we also carry out a seasonal analysis. The results of this are shown in Figure D1 and Figure D2 in boxplot form (see annex). Table E1 and Table E2 list all performance metrics for the entire study period as well as seasonally. For the entire 2018–2019 period, R is very similar for both datasets with R=0.93 for the downscaled product and R=0.92 for ERA5-Land. In comparison to ERA5-Land, the downscaled product has a RMSE of 22.53 vs 25.7 $W^2$. The average bias is lower for ERA5-Land, with -1.56 vs -6.83 $W^2$.

The downscaled product shows a better performance for the summer period AMJ and JAS (R=0.91 and 0.93 vs 0.83 and 0.86) and the same is true in terms of RMSE (27.58 and 22.18 $W^2$ vs 34.79, 29.37 $W^2$). The seasonal bias is lower for the downscaled product.

Figure 12 shows as an example the SNR for the downscaled product and ERA5-Land for the 30th of June over an area of western Europe. The increase in spatial resolution and therefore landscape details is clearly visible. The downscaled dataset both shows higher and lower values than ERA5-Land as it is able to resolve finer land surface features due to the high-resolution merged LST and Albedo inputs.

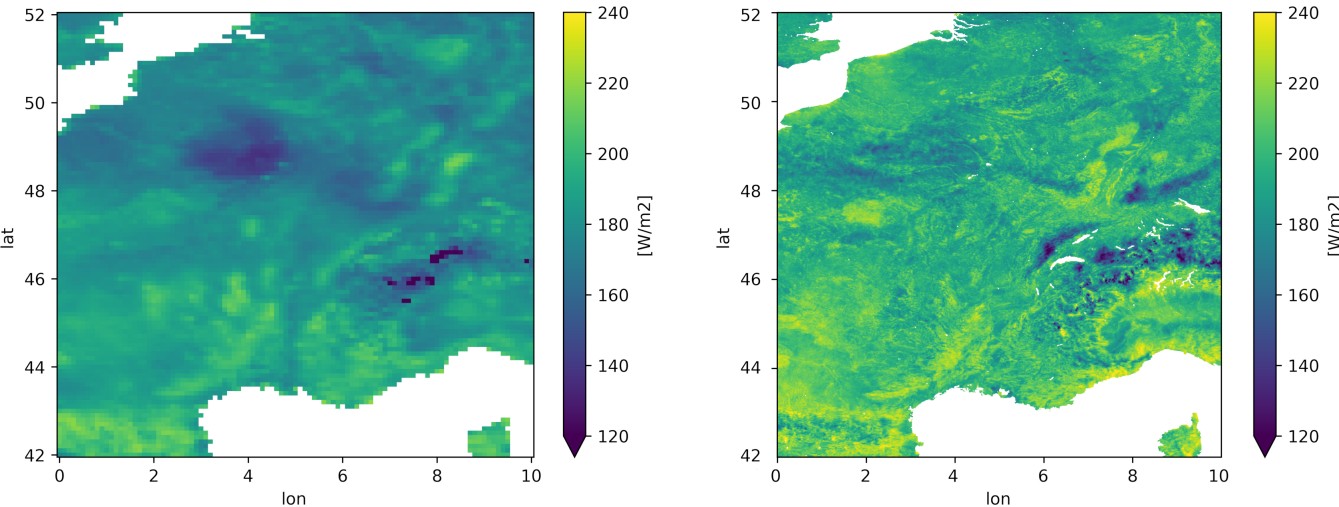

**Figure 12.** SNR from ERA5-Land (left) and the downscaled dataset (right) for 30th June 2018. The shown maps depict a large part of western Europe covering France, Germany and Italy. Data gaps around lakes and shorelines due to the relatively coarser resolution of the LSAF inputs have been filled through bilinear interpolation and a 1 km water mask has been applied.

## 5 Discussion

The methodology described and validated above to produce gap-free all-sky SNR at 1 km resolution relies on producing gap-free 1 km $SW_{out}$ and $LW_{out}$ estimates. The methodology to produce $SW_{out}$, namely by means of bias-correction, is relatively straightforward. The more complex multi-stage approach taken for the all-sky LST estimates, required to compute $LW_{out}$, is
355 discussed in some more detail here with regards to similar studies. Some further remarks on other available SNR products, and the comparison of the here created SNR product to ERA5-Land, as well as the validation of the individual radiation components, follow.

Examples of other gap-free LST datsets which have recently been developed are given by Shiff et al. (2021), Xu and Cheng
(2021), Jia et al. (2022) or Wu et al. (2023). The approach taken by Shiff et al. (2021) was to merge clear-sky 1 km MODIS LST with 0.2 degree modelled air temperature provided by the National Center for Environmental Prediction (NCEP) from the Coupled Forecast System Model version 2 (CFSv2) system. This was done by extracting the underlying seasonal behaviour from both input datasets by Temporal Fourier Analysis and subsequently adding the CSFv2 anomalies to the MODIS climatology on days where no clear-sky MODIS LST observation is available. Xu and Cheng (2021) demonstrated a multi-step approach
based on infrared Advanced Microwave Scanning Radiometer 2 (AMSR2) brightness temperatures, MODIS LST as well as MODIS based ancillary datasets and elevation data. First, land surface temperature is retrieved from all the above datasets at 0.1 degree spatial resolution. This LST dataset is then downscaled from 0.1 degree resolution to 0.01 degree resolution by using the elevation data and MODIS NDVI. Clear-sky MODIS LST data and the retrieved all-sky LST data are then bias corrected

allowing for the temporal gap-filling of the clear-sky LST retrievals. Finally, the 0.1 degree LST retrievals based on AMSR2 are

assimilated into the merged 0.01 degree LST dataset by applying a multiresolution Kalman filtering approach. Jia et al. (2022) have produced all-sky diurnal, hourly LST estimates at 2 km spatial resolution based on the surface energy balance. The three step approach is based on constructing a spatiotemporal dynamic model of LST from ERA-5 in which clear-sky LST from the Advanced Baseline Imager (ABI) are assimilated. As a final step the gap-free LST record is updated by superimposing diurnal cloud effects using satellite radiation products. Wu et al. (2023) have tested an approach to produce very high-resolution, 100m

gap-free LST, from a single Landsat-8 acquisition by training a Random Forest algorithm with the Landsat derived LST and ancillary variables, e.g. land cover, population density and elevation. The LST merging methodology presented in this paper shares some of the elements of the above mentioned studies, i.e. primarily the bias-correction of the coarse-scale LSAF LST observations towards Sentinel 3 (see section 3.3), as well as a Kalman Filtering approach. An in-depth validation and quantitative inter-comparison of the above mentioned products was not the aim of this study presented here. We argue however, that

on a theoretical basis, the here proposed methodology has some advantages. Most of the above mentioned approaches rely on input data with a coarser spatial resolution. Shiff et al. (2021) and Jia et al. (2022) for instance use air temperature data at a 0.2 degree resolution or ERA-5 with approximately 31 km spatial resolution. Both these datasets are also model output, albeit from data assimilation systems taking a multitude of observations into account. The coarsest spatial resolution of the input datasets used in the here presented methodology are the LSAF geostationary retrievals with a pixel size of ca. 5-7 km, depend-

ing on latitude. While especially the LSAF all-sky retrievals also are based on modelling, and require ancillary information, they are optimised for the retrieval of the single target variable at high accuracy. They are also available hourly, like ERA-5, whereas e.g. Landsat-8, used by Wu et al. (2023), is only available every few days, depending on cloud cover. Furthermore, our approach does not rely on using ancillary variables which are not directly linked to the physical processes to statistically downscale the input products, as is for example done in Wu et al. (2023) by using population density. One of the drawbacks

of the here presented methodology is the lack of a dynamic temporal model which is able to propagate assimilation updates, provided by the 1 km LST retrievals from Sentinel-3, over time, which has been achieved by Jia et al. (2022). Here we thus apply the same update from when a Sentinel-3 observation is available to the subsequent time steps until the next Sentinel-3 observation is available. An additional drawback is that e.g. ERA-5 and NCEP are globally available datasets and the use of MODIS LST retrievals allows for the production of long time series. In contrast, LSAF is limited to North Africa and Europe

and Sentinel-3 was only launched in 2016. The approach can however be transferred to other regions by substituting LSAF with other geostationary retrievals and using MODIS instead or in addition to Sentinel-3 to allow for an extension of the time series.

In terms of the calculation of the daily all-sky surface net radiation dataset, we argue that the approach taken is the most

straightforward as it is based on the underlying physical principles of the individual radiation components. This is in contrast to studies presenting methods to produce net radiation at a similar temporal and spatial resolution which exploit statistical relationships between some well observed components, e.g. incoming radiation components from satellite, and ancillary information, e.g. land cover or NDVI, or modelled variables. Xu et al. (2022) for example train a convolutional neural network

using net radiation from a selection of in-situ measurements, MERRA-2 reanalysis and AVHRR top of atmosphere (TOA) data.

Jiang et al. (2023) presented two algorithms based on a Random Forest to downscale the GLASS net radiation product, either by exploiting the relationship between net radiation and shortwave radiation as well as ancillary information, including from ground measurements, or by linking net radiation to TOA observations from the Landsat satellites and ancillary information. The GLASS algorithm itself is based on the Multivariate Adaptive Regression Splines (MARS) model, trained with remotely sensed incoming radiation, NDVI and albedo as well as mostly MERRA-2 meteorological variables (Jiang et al., 2016). While such downscaling methodologies can work very well, and we need to note that no quantitative comparison is here performed, they rely on training a model which establishes a statistical relationship between the different input variables. These data driven approaches are very sensitive to the training data and e.g. the spatial or temporal domain for which such a model is established. Hence, a globally trained model might not capture locally specific conditions or provide accurate output for time periods not considered for the training. With in-situ training data often the limiting factor, established statistical relationships might also be only valid for these specific sites and avoiding model over-fitting can be very challenging. It can thus be beneficial if in-situ measurements are solely used for the validation of a methodology rather than the development itself. Another methodology to produce hourly surface solar radiation at 5 km spatial resolution was developed by Tang et al. (2016). In the two step approach hourly cloud parameters are estimated with a neural network by combining cloud products from MODIS with high temporal-resolution top-of-the-atmosphere (TOA) radiance data from the geostationary Multifunctional Transport Satellite (MTSAT). Subsequently the cloud information and other auxiliary information is combined in a radiative transfer model to retrieve the surface net radiation. Conceptually, although estimating surface radiation primarily based on cloud properties, is is similar to the here presented approach in exploiting the advantages of geostationary and polar-orbiting satellite measurements and being more physically based. An overview of some further approaches to produce surface net radiation products are also given by Tang et al. (2016).

We need to state that in this paper we make no accuracy comparisons between the different approaches mentioned above. Also, in terms of ancillary variables, this study indirectly relies on these through the use of the chosen input products. The retrieval of LST for instance, especially in cloudy conditions, relies on modelled processes requiring information such as vegetation phenology. In terms of the validation of the produced SNR product and the individual radiation components, we also acknowledge that the in-situ measurements have an error (difference to the 'truth' at the local scale that they sample), but that the pixel-to-local representativeness error, i.e. the difference between the pixel truth that we aim for, and the local truth at the smaller tower footprint, is much larger. Unfortunately, we cannot solve this issue but argue that using as many stations as possible benefits the validation, particularly within pixels where the spatial heterogeneity is very large. Finally, by relying on input products directly representing components of the surface radiation balance, any future enhancements in the source products should directly lead to improvements in future releases of the 1km SNR dataset.

# 6 Conclusions

Surface net radiation is a key input variables for many land surface and hydrological models. With increased efforts to simulate land surface processes at higher spatial resolution, the lack of high-resolution gap-free SNR is an issue. Heterogeneity of model output is then primarily driven by land surface properties for which high-resolution datasets are more frequently available (e.g. soil texture, vegetation phenology). In this paper we presented a methodology to systematically combine the advantages of frequent geostationary LST and radiation observations, enhanced with modelled data when cloud cover inhibits the direct retrieval, with LST and albedo retrievals from polar-orbiting satellites at high spatial resolution. The resulting gap-free net radiation dataset, as well as the intermediate all-sky LST dataset, for 2018–2019 across Europe uses operationally available input datasets which opens up the possibility to update the data on a close to near-real time basis. Based on the surface energy balance, and optimising each radiation component individually using input datasets with already a high accuracy, some improvements are achieved in addition to a substantial increase in spatial heterogeneity and representativeness.

While a gap-free LST datset was developed within this study, the validation of the dataset was carried out indirectly based on $LW_{out}$ measurements. This served the purpose of the study to ultimately create a SNR dataset.

Conceptually, one of the advantages of the here developed LST merging methodology within the overall scope of producing net radiation, is its reliance on one of the LST input products being provided by LSAF, thus making the approach more consistent as the incoming radiation components are also LSAF products. The use of Sentinel 3 SLSTR emissivity maps when computing the outgoing longwave radiation $LW_{out}$ should be considered in future product updates to make the methodology even more consistent. In addition, the presented results are based on the use of LST retrievals from the Sentinel 3A satellite and data from Sentinel 3B should be incorporated in the future. Also, the use of Sentinel-3 based albedo instead of PROBA-V should be explored. A limitation in the downscaling methodology is that in the assimilation step, performed after the bias correction of LSAF LST towards Sentinel-3, there is no dynamic model to propagate the updates from the Sentinel 3 LST assimilation at the daytime or nighttime overpass time to the subsequent hours. To paliate this issue, we applied equivalent updates to the subsequent hourly LSAF observations, separately for temporal daytime/nighttime windows. Alternative approaches – such as the attenuation of the assimilation impact over time – could be explored based on a more in-depth analysis of the diurnal cycle. While the validation presented concentrated on daily aggregates, the availability of hourly LST and radiation products does make it possible to resolve the diurnal cycle, which can be a requirement for certain models.

In principle the approach developed within this study can be extended to other areas where there are both geostationary and polar-orbiting observations, not necessarily the ones used for this study. The here presented dataset shall be updated in the future as we consider it to be an ideal input dataset for high-resolution land surface applications, e.g. for the Global Land Evaporation Amsterdam Model (Martens et al., 2017).

# 7 Data availability

The daily SNR and LST datasets for 2018–2019 are available for scientific use under https://doi.org/10.5281/zenodo.8332222 / https://doi.org/10.5281/zenodo.8332128 as netcdf files (RNETdaily_lon_lat.nc and LSTdaily_lon_lat.nc), see Rains (2023a) and Rains (2023b). The spatial domain covered by the product is -11.5 to 26.5 longitude and 35 to 71 latitude.

## Appendix A: In-situ sites

| ID | name | lon | lat | IGBP | SW_in | LW_in | SW_out | LW_out |
|---|---|---|---|---|---|---|---|---|
| BE-Dor | Dorinne | 4.968 | 50.312 | GRA | x | x | x | x |
| BE-Lcr | Lochristi | 3.850 | 51.112 | DBF | x | x | x | x |
| BE-Lon | Lonzee | 4.746 | 50.552 | CRO | x | x | x | x |
| BE-Maa | Maasmechelen | 5.632 | 50.980 | CSH | x | x | x | x |
| BE-Vie | Vielsalm | 5.998 | 50.305 | MF | x | x | x | x |
| CH-Aws | Alp Weissenstein | 9.790 | 46.583 | GRA | x | x | | |
| CH-Cha | Chamau | 8.410 | 47.210 | GRA | x | x | x | x |
| CH-Dav | Davos | 9.856 | 46.815 | ENF | x | x | x | x |
| CH-Fru | Früebüel | 8.538 | 47.116 | GRA | x | x | | |
| CH-Lae | Laegern | 8.364 | 47.478 | MF | x | x | | |
| CH-Oe2 | Oensingen | 7.734 | 47.286 | CRO | x | x | | |
| CZ-Lnz | Lanzhot | 16.946 | 48.682 | MF | x | x | x | x |
| CZ-RAJ | Rajec | 16.697 | 49.444 | ENF | x | x | x | x |
| CZ-Stn | Stitna | 17.970 | 49.036 | DBF | x | x | x | x |
| CZ-Wet | Trebon | 14.770 | 49.025 | WET | x | x | x | x |
| DE-Akm | Anklam | 13.683 | 53.866 | WET | x | x | x | x |
| DE-Dgw | Dagowsee | 13.054 | 53.151 | WET | x | x | x | x |
| DE-Geb | Gebesee | 10.915 | 51.100 | CRO | x | x | x | x |
| DE-Gri | Grillenburg | 13.513 | 50.950 | GRA | x | x | x | x |
| DE-Hai | Hainich | 10.452 | 51.079 | DBF | x | x | x | x |
| DE-HoH | Hohes Holz | 11.219 | 52.085 | DBF | x | x | x | x |
| DE-Hte | Huetelmoor | 12.176 | 54.210 | WET | x | | | |
| DE-Hzd | Hetzdorf | 13.490 | 50.964 | DBF | x | x | x | x |
| DE-Kli | Klingenberg | 13.522 | 50.893 | CRO | x | x | x | x |
| DE-Obe | Oberbärenburg | 13.721 | 50.787 | ENF | x | x | x | x |
| DE-RuR | Rollesbroich | 6.304 | 50.622 | GRA | x | x | x | x |
| DE-RuS | Selhausen Juelich | 6.447 | 50.866 | CRO | x | x | x | x |
| DE-Tha | Tharandt | 13.565 | 50.963 | ENF | x | x | x | x |

| | | | | | | | | |
|---|---|---|---|---|---|---|---|---|
| DE-Zrk | Zarnekow | 12.889 | 53.876 | WET | x | x | x | x |
| DK-Sor | Soroe | 11.645 | 55.486 | DBF | x | x | x | x |
| ES-Abr | Albuera | -6.786 | 38.702 | SAV | x | x | x | x |
| ES-Cnd | Conde | -3.228 | 37.915 | WSA | x | x | x | x |
| ES-LM1 | Majadas del Tietar North | -5.779 | 39.943 | SAV | x | x | x | x |
| ES-LM2 | Majadas del Tietar South | -5.776 | 39.935 | SAV | x | x | x | x |
| FI-Hyy | Hyytiala | 24.295 | 61.847 | ENF | x | x | x | x |
| FI-Kmp | Kumpula | 24.961 | 60.203 | URB | x | x | x | x |
| FI-Kvr | Kuivajarvi | 24.280 | 61.847 | WAT | x | x | x | x |
| FI-Let | Lettosuo | 23.960 | 60.642 | ENF | x | x | x | x |
| FI-Sii | Siikaneva | 24.193 | 61.833 | WET | x | x | x | x |
| FI-Var | Varrio | 29.610 | 67.755 | ENF | x | | | |
| FR-Aur | Aurade | 1.106 | 43.550 | CRO | x | x | x | x |
| FR-Bil | Bilos | -0.956 | 44.494 | ENF | x | x | x | x |
| FR-EM2 | Estrees-Mons A28 | 3.021 | 49.872 | CRO | x | x | | |
| FR-FBn | Font-Blanche | 5.679 | 43.241 | MF | x | x | x | x |
| FR-Fon | Fontainebleau | 2.780 | 48.476 | DBF | x | x | x | x |
| FR-Gri | Grignon | 1.952 | 48.844 | CRO | x | x | x | x |
| FR-Hes | Hesse | 7.065 | 48.674 | DBF | x | x | x | x |
| FR-LGt | La Guette | 2.284 | 47.323 | WET | x | x | x | x |
| FR-Mej | Mejusseaume | -1.796 | 48.118 | GRA | x | x | x | x |
| FR-Pue | Puechabon | 3.596 | 43.741 | EBF | x | x | x | x |
| IT-BCi | Borgo Cioffi | 14.957 | 40.524 | CRO | x | x | x | x |
| IT-Cp2 | Castelporziano2 | 12.357 | 41.704 | EBF | x | x | | x |
| IT-Lsn | Lison | 12.750 | 45.740 | OSH | x | x | x | x |
| IT-MtM | Muntatschinig Meadow | 10.580 | 46.687 | GRA | x | x | x | x |
| IT-Ren | Renon | 11.434 | 46.587 | ENF | x | | x | |
| IT-SR2 | San Rossore 2 | 10.291 | 43.732 | ENF | x | x | x | x |
| IT-Tor | Torgnon | 7.578 | 45.844 | GRA | x | x | x | x |
| RU-Fy2 | Fyodorovskoye | 32.902 | 56.448 | ENF | x | x | x | x |
| RU-Fyo | Fyodorovskoye | 32.922 | 56.462 | ENF | x | x | x | x |
| SE-Deg | Degero | 19.557 | 64.182 | WET | x | x | x | x |
| SE-Htm | Hyltemossa | 13.419 | 56.098 | ENF | x | x | x | x |
| SE-Lnn | Lanna | 13.102 | 58.341 | CRO | x | | | |
| SE-Nor | Norunda | 17.480 | 60.086 | ENF | x | x | x | x |
| SE-Svb | Svartberget | 19.775 | 64.256 | ENF | x | x | x | x |
| bud | Budapest-Lorinc | 19.182 | 47.429 | | x | x | x | x |
| cab | Cabauw | 4.927 | 51.971 | | x | x | x | x |

| car | Carpentras | 5.030 | 44.050 | x | x | | |
| cnr | Cener | -1.601 | 42.816 | x | x | | |
| lin | Lindenberg | 14.122 | 52.210 | x | x | | |
| pal | Palaiseau | 2.208 | 48.713 | x | x | | |
| pay | Payerne | 6.944 | 46.815 | x | x | x | x |
| son | Sonnblick | 12.958 | 47.054 | x | x | | |
| tor | Toravere | 26.462 | 58.264 | x | x | x | x |

## Appendix B: Incoming radiation fluxes

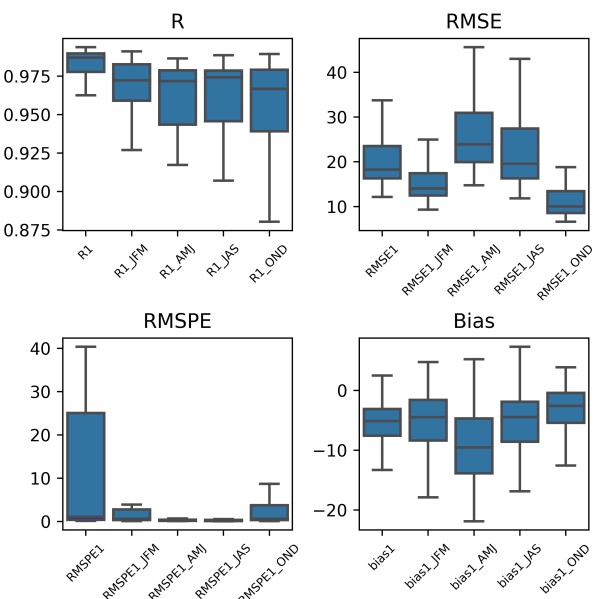

**Figure B1.** Validation of LSAF $SW_{in}$ in terms of R, RMSE, RMSPE and bias for the entire period as well as seasonally.

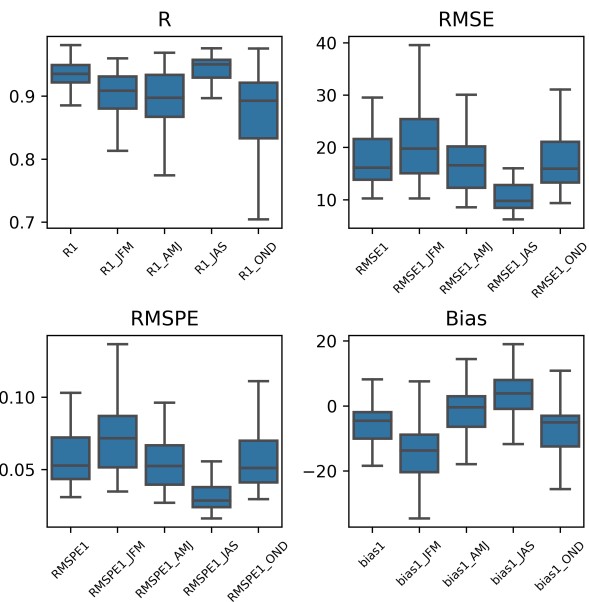

**Figure B2.** Validation of LSAF $LW_{in}$ in terms of R, RMSE, RMSPE and bias for the entire period as well as seasonally.

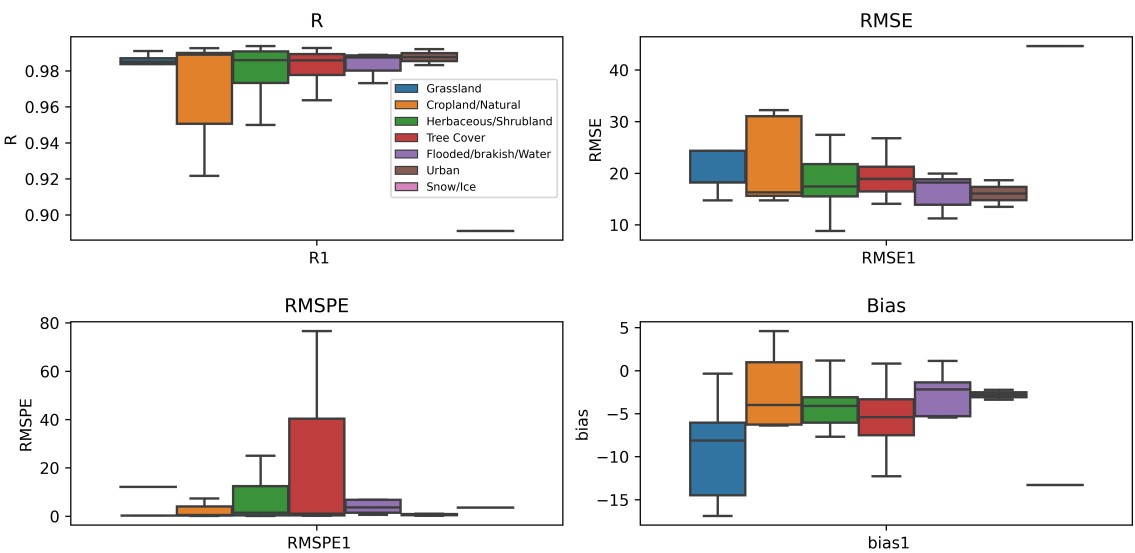

**Figure B3.** Validation of LSAF $SW_{in}$ in terms of R, RMSE, RMSPE and bias for different land cover types.

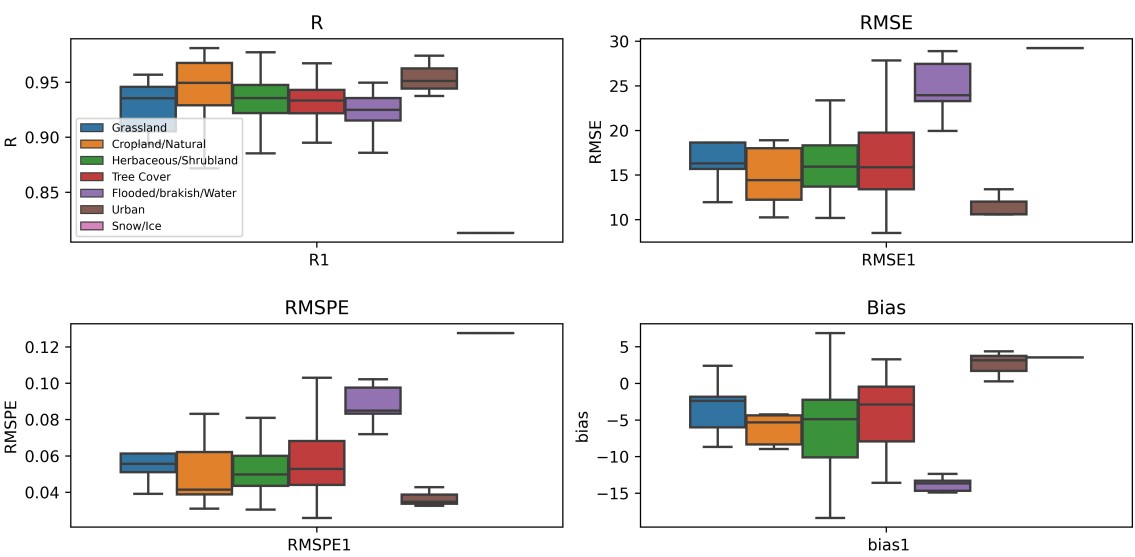

**Figure B4.** Validation of LSAF $LW_{in}$ in terms of R, RMSE, RMSEP and bias for different land cover types.

## Appendix C: Outgoing radiation fluxes

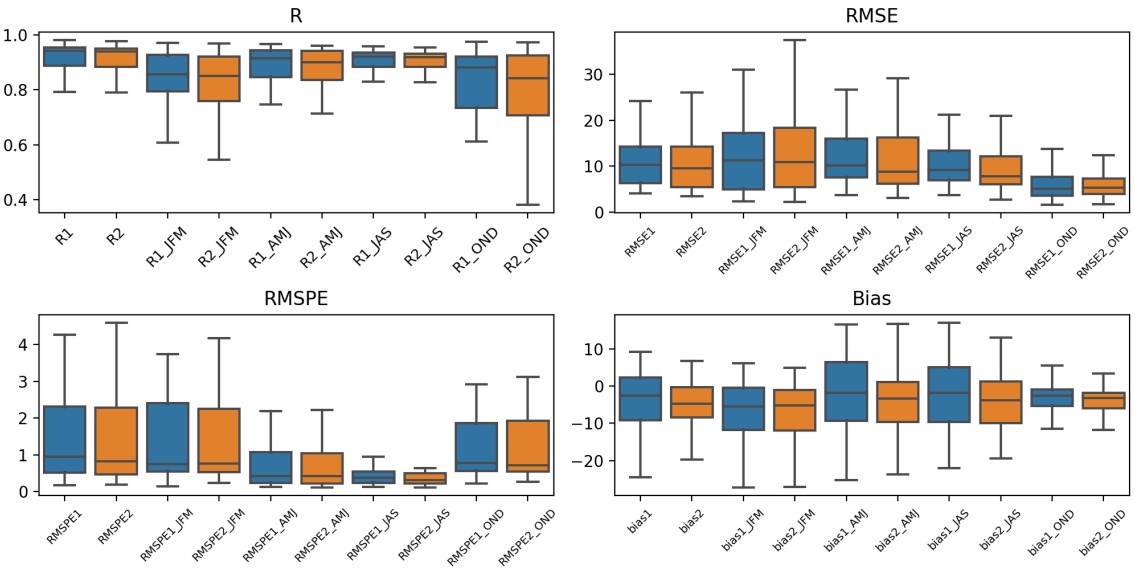

**Figure C1.** Validation of $SW_{out}$ in terms of R, RMSE, RMSPE and bias using LSAF only (R1) and the downscaled product (R2) for the entire period as well as seasonally.

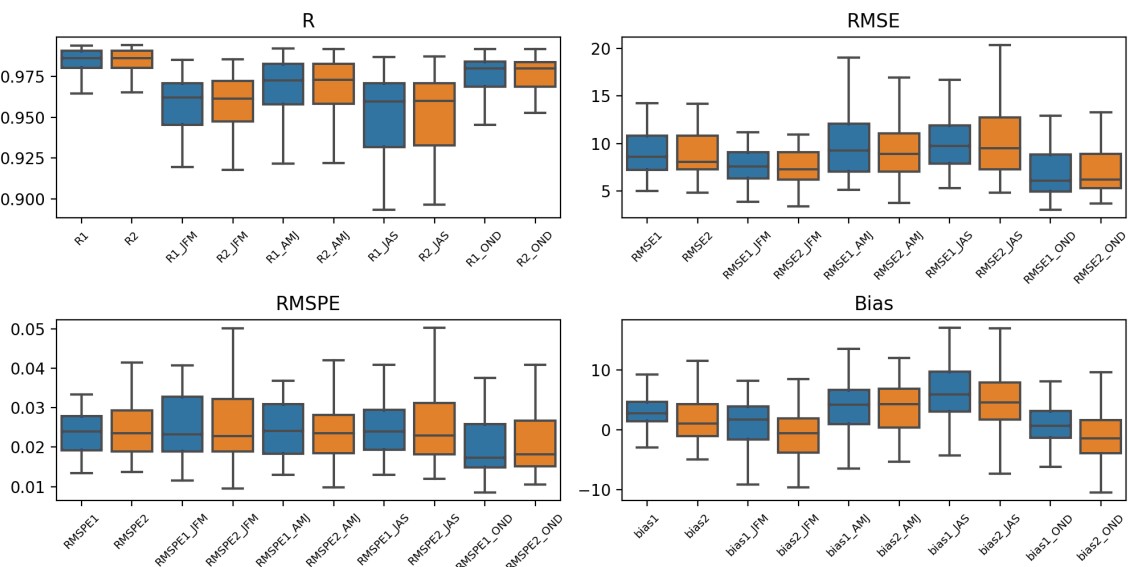

**Figure C2.** Validation of $LW_{out}$ in terms of R, RMSE, RMSPE and bias using LSAF only (R1) and the the downscaled product (R2) for the entire period as well as seasonally.

## Appendix D: Net radiation

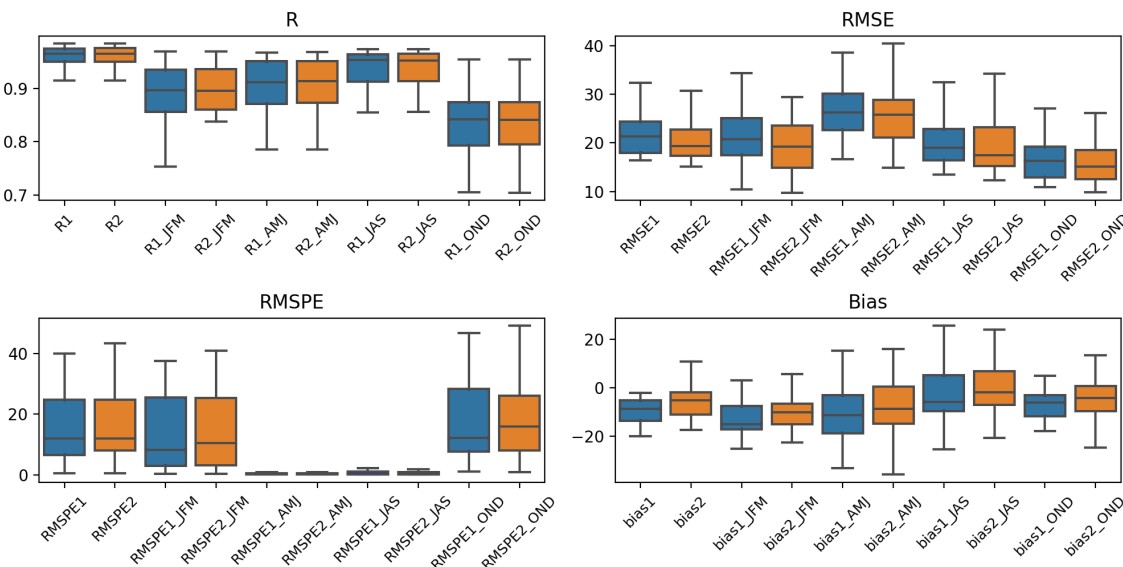

**Figure D1.** Validation of SNR in terms of R, RMSE, RMSPE and bias using LSAF only (R1) and the downscaled product (R2) for the entire period as well as seasonally.

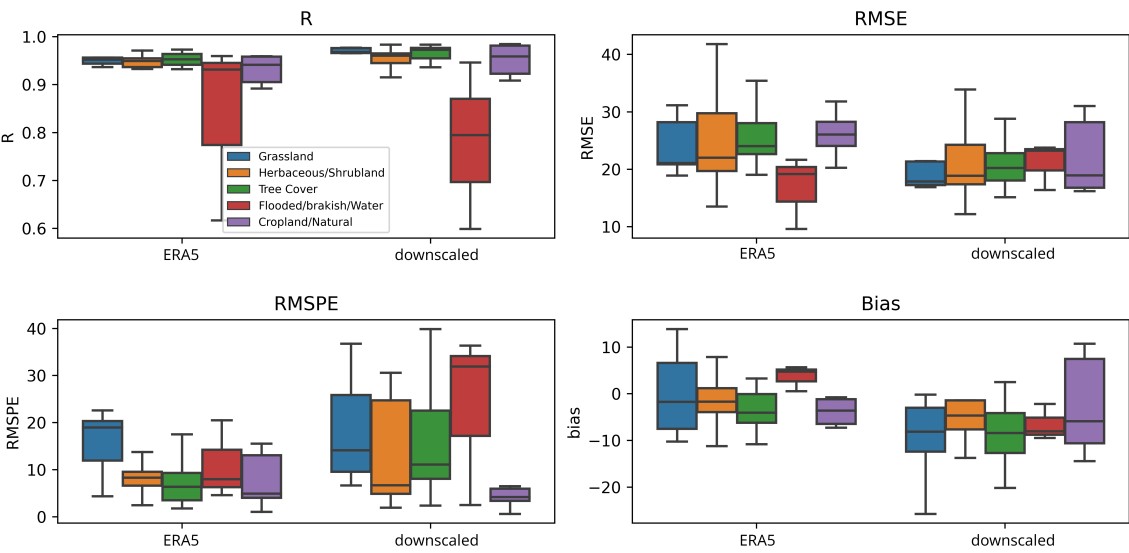

**Figure D2.** Validation of ERA5-Land and downscaled net radiation product against in-situ measurements in terms of R, RMSE, RMSEP and bias for different land cover types.

**Appendix E:  Overall validation statistics**

|  | R | MSE | MSPE | bias |
|---|---|---|---|---|
| $SWin$ | 0.97 | 876 | 7.59 | -7.65 |
| $LWin$ | 0.93 | 420 | 0.06 | -6.99 |
| $SWout$ LSAF | 0.87 | 317 | 7.99 | -4.55 |
| $SWout$ RADLST | 0.87 | 293 | 6.93 | -5.5 |
| $LWout$ LSAF | 0.97 | 132 | 0.029 | 2.36 |
| $LWout$ RADLST | 0.97 | 122 | 0.028 | 0.81 |
| RNET LSAF | 0.93 | 551 | 17 | -9.06 |
| RNET RADLST | 0.93 | 515 | 15.89 | -6.11 |
| ERA5 | 0.93 | 654 | 10.04 | -1.89 |

**Table E1.** Performance metrics for radiation components for the 2018–2019 study period.

| | R Q1 | MSE Q1 | MSPE Q1 | bias Q1 | R Q2 | MSE Q2 | MSPE Q2 | bias Q2 | R Q3 | MSE Q3 | MSPE Q3 | bias Q3 | R Q4 | MSE Q4 | MSPE Q4 | bias Q4 |
|---|---|---|---|---|---|---|---|---|---|---|---|---|---|---|---|---|
| $SWin$ | 0.96 | 535 | 6.66 | -6.88 | 0.95 | 1320 | 1.09 | -12.2 | 0.94 | 1431 | 1.91 | -8.02 | 0.95 | 399 | 5.97 | -4.92 |
| $LWin$ | 0.89 | 598 | 0.08 | -15 | 0.88 | 361 | 0.05 | -2.39 | 0.93 | 228 | 0.03 | 1.05 | 0.87 | 433 | 0.06 | -9.29 |
| $SWout$ LSAF | 0.84 | 588 | 11.5 | -9.14 | 0.87 | 490 | 1.99 | -3.26 | 0.89 | 148 | 4.63 | -2.72 | 0.8 | 129 | 3.36 | -4.01 |
| $SWout$ RADLST | 0.82 | 562 | 9.25 | -9.66 | 0.87 | 441 | 1.66 | -5.07 | 0.89 | 121 | 3.87 | -4.23 | 0.78 | 124 | 3.22 | -4.32 |
| $LWout$ LSAF | 0.92 | 114 | 0.029 | 0.87 | 0.94 | 170 | 0.03 | 4.31 | 0.92 | 145 | 0.02 | 5.94 | 0.95 | 96 | 0.02 | 0.27 |
| $LWout$ RADLST | 0.93 | 101 | 0.028 | -0.83 | 0.95 | 163 | 0.03 | 2.87 | 0.93 | 134 | 0.02 | 4.6 | 0.96 | 90 | 0.02 | -1.36 |
| RNET LSAF | 0.84 | 527 | 21 | -11.87 | 0.91 | 860 | 1.11 | -10.55 | 0.93 | 503 | 1.96 | -4.17 | 0.77 | 336 | 22.24 | -7.82 |
| RNET RADLST | 0.84 | 481 | 20 | -9.39 | 0.91 | 800 | 1.14 | -6.6 | 0.93 | 477 | 2.03 | -0.76 | 0.8 | 316 | 19.79 | -5.51 |
| ERA5 | 0.84 | 407 | 10.49 | 47 | 0.83 | 1187 | 1.07 | -61 | 0.86 | 844 | 2.43 | -50 | 0.82 | 274 | 13.86 | 53.46 |

**Table E2.** Seasonal performance metrics for radiation components.

## Appendix F:  Downscaling of LSAF LST with Sentinel 3 LST

For the downscaling/merging of the LSAF with Sentinel 3 based LST retrievals described in section 3.3 some more detail is given here. Figure F1 shows as an example the mean Sentinel 3 LST and its bias towards LSAF observations for daytime (10am. local time) observations. Across the domain the bias is neither systematically negative nor positive, highlighting the generally high agreement between LSAF and Sentinel 3 observations, and it is more linked to geographic features. The UTC time of the underlying Sentinel 3 data is different for each pixel/day across the domain and the LSAF data the bias is calculated against is thus a composite from different acquisition times. The Sentinel 3 observations are normalised to the one the hour Sentinel 3 mean overpass time per pixel to enable a more correct match-up between Sentinel 3 and LSAF (as the LSAF data is representative for on the hour). This is done through linear interpolation using the LSAF LST difference between the full hour before and after the exact overpass time of each Sentinel 3 observation. The bias correction is then performed between LSAF LST and the normalised Sentinel 3 observations for each pixel individually for the entire study period. A seasonal bias correction should be considered in the future.

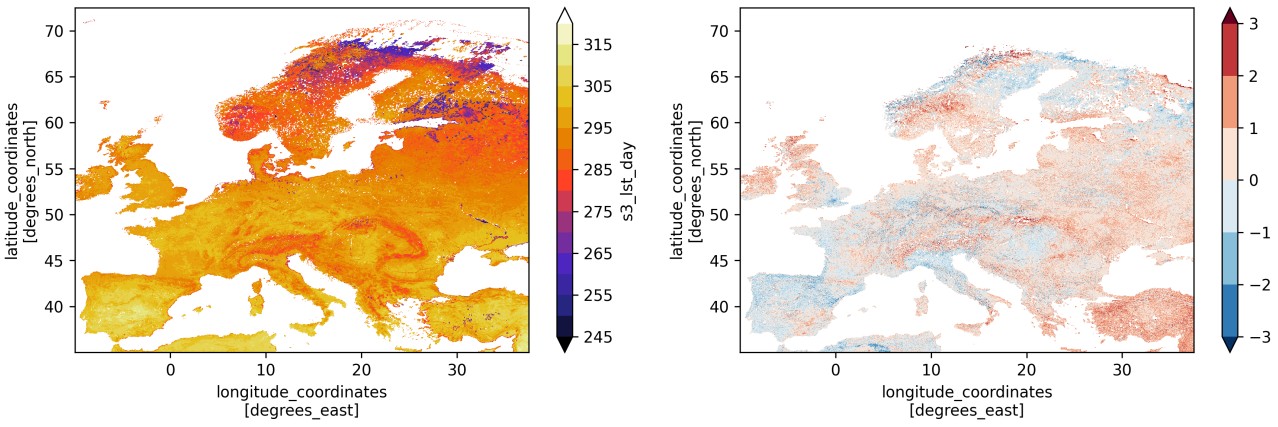

**Figure F1.** Mean LST of Sentinel 3 daytime (ca. 10am) observations (left) and bias towards LSAF observations (right).

After the full bias correction of the hourly LSAF data the normalised Sentinel 3 observations are assimilated into this time series for each pixel. The respective uncertainties of both Sentinel 3 and LSAF LST retrievals for each pixel/timestep are therefore taken into account. Figure F2 shows as an example for a single day the assimilation diagnostics. The top row shows the Sentinel 3 LST retrieval (left), the uncertainty map of the Sentinel 3 observation (middle) and the uncertainty of the LSAF observations (right). The Kalman Gain (bottom left) is based on the two uncertainties and a value of 1 would fully trust the Sentinel 3 observation, whereas 0 would result in no assimilation update. The difference, i.e. innovation, between the Sentinel 3 observation and LSAF LST, is shown in the lower middle. The increment, the actual update, is the innovation multiplied by the Kalman Gain and is shown in the bottom right.

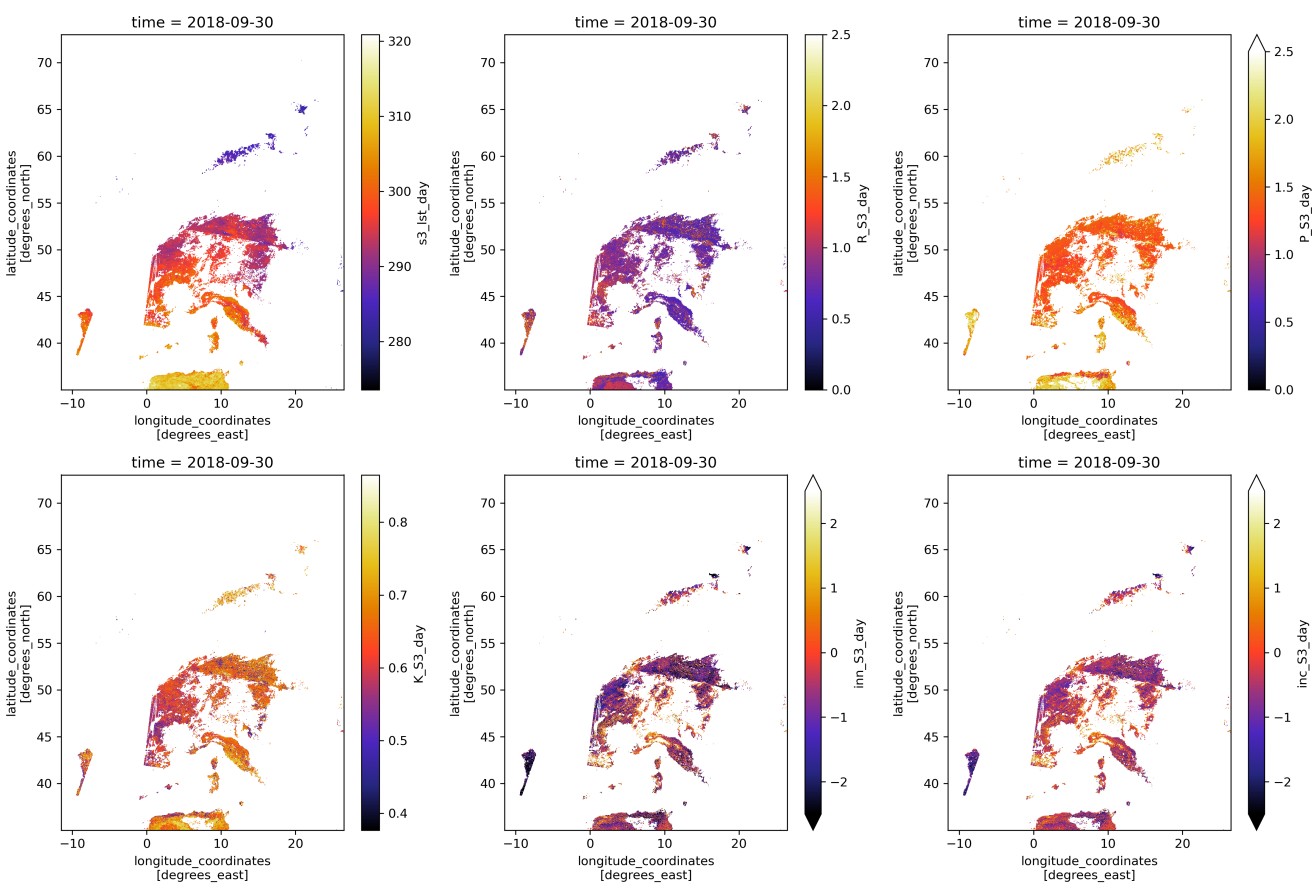

**Figure F2.** Sentinel 3 LST retrievals (top left), uncertainty of Sentinel 3 LST retrievals (top middle), uncertainty of LSAF LST retrievals (top right) and Kalman Gain (bottom left), innovations (bottom middle), increments (bottom right).

Figure F3 shows the 2018–2019 mean assimilation diagnostics for the daytime Sentinel 3 assimilation. The innovation (left) is fairly close to zero showing that the bias correction results in the Sentinel 3 observations being on average spread evenly around the bias corrected LSAF time series as intended. The mean increment (middle), the actual correction applied to the LSAF estimates, shows similar spatial patterns. The mean Kalman Gain is shown on the right.

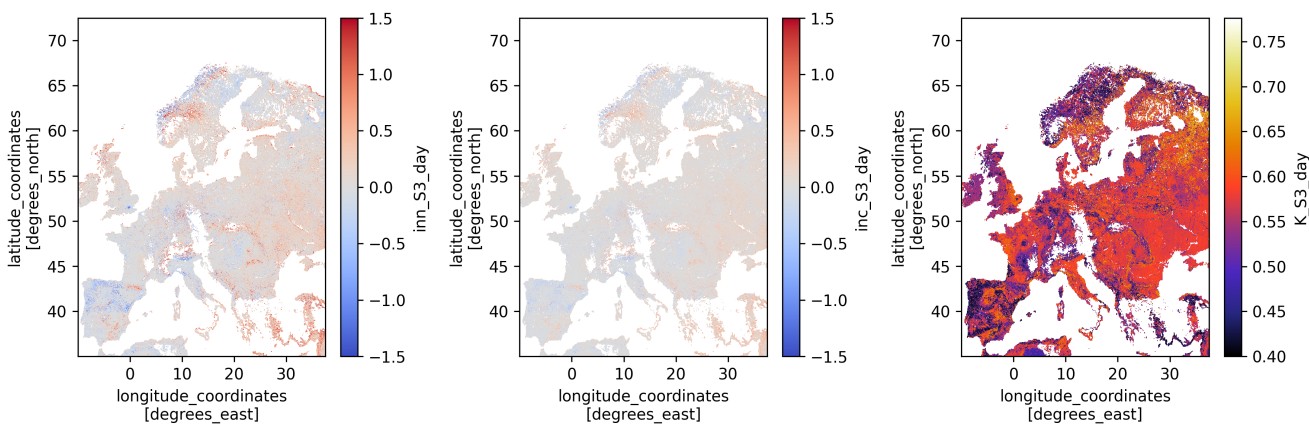

**Figure F3.** Mean Innovation (left), increments (middle) and Kalman Gain (right) for daytime Sentinel 3 LST assimilation.

*Competing interests.* No competing interests are present.

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
