# Peer review of "High-resolution (1-km) all-sky net radiation over Europe enabled by the merging of land surface temperature retrievals from geostationary and polar-orbiting satellites."

_Earth System Science Data, 2022_

## Author Comment (AC1)

This manuscript introduced a downscaled and continuous daily LST and SNR product across Europe for 2018–2019. The validations of radiation against BSRN in-situ measured are also presented in the paper. And it is said that an improvement of the root mean squared error by ca.8% with a substantial increase in spatial detail compared to the original MSG product. The paper indicates that the resulting pan-European LST and SNR dataset can be used for hydrological modelling and as input to models dedicated to estimating evaporation and surface turbulent heat fluxes. The LST and SNR product is important to describe Earth surface energy balance. Overall, this manuscript is clear. And the study is of great significance to improve the new understanding of energy balance in Europe. However, there are several issues that need to be taken care of before this paper becomes acceptable for publication.

We thank the reviewer for the time and effort put into this review. Both the general comment above and the detailed comments below are very helpful for the improvement of this manuscript. We are happy to share our point-by-point reposes highlighted in blue.

1. the high resolution LST product is merged from LEAF (all sky) and Sentinel 3 LST (clear sky). The two LSTs have different spatial and temporal resolutions. While doing the merging, if any cloud effect is considered? If any cloud product is involved? If yes, please indicated it.

   Response:

   We have not included any cloud product although the prevalence of clouds is considered in the merging procedure. Cloud cover is assumed when no LSAF clear-sky data is available and we fall back to the all-sky dataset. Therefore:

   1) Cloud over is taken into account by only using clear-sky LSAF data for the computation of the bias between LSAF and Sentinel-3, both daytime and nighttime.

   2) When computing the diurnal cycle effect on the misalignment of the Sentinel-3 observation to the full hour, this is only done if the two required LSAF samples on which the linear-interpolation is performed are clear-sky observations.

   3) In the Kalman assimilation scheme a higher uncertainty is assumed for LSAF all-sky products than for the clear-sky equivalent.

   Action: We will clarify this in the revise manuscript by providing more details in Sect 3.3 on page 7 as well as in the discussion and appendix.

2. while downscaling the LST product, if any edge effects (coast lines, cloud edges) are considered?

Response:

Due to the coarser spatial resolution of LSAF coast lines are not as well represented as in the 1 km Sentinel-3 data, as  pixels with a considerable amount of water are masked  out. The merging procedure and calculation of all-sky LST and net radiation at 1 km resolution can only be carried out where all input datasets are available and therefore the same rugged coast lines are visible in the LST/net radiation product.

Action: A simple solution, albeit introducing some uncertainties, is to extrapolate from the closest pixels with valid data on a daily basis. We will implement this in the revised manuscript and make it clear which pixels are based on interpolation through a mask value.

3. Line 220, it is said "Extensive validation of the LSAF and Sentinel 3 LST products has already been performed (see below). Both have an average accuracy below 1.5 K, although it varies across space and time. Our goal is to combine their individual strengths in terms of spatial and temporal resolution to obtain an enhanced representation of landscape heterogeneity". Although there are extensive validations of the LSAF and Sentinel 3 LST products, the validations are based on different spatial and temporal resolutions. It does not mean that the merged product could also has a good performance. It is good to give the statistics.

   Response: We agree with the reviewer.

   Action: We will include more detailed validation statistics in Sect 4.2 (see also the next comment given by the reviewer).

4. The paper is lack of statistics. e.g. figure 1, any overall statistics could be summarized in a table? And the absolute RMSEs are given in Figure 1. The percentage-wise is worth known. And so does the validations of outgoing raditaions and SNR. Please summarize the overall statistics (R, bias, RMSE (including percentages)), degree of freedom) in tables.

   Response: We agree that they should be included.

   Action: We will include all the above mentioned statistics and seasonal analysis will be included as well in Sect 4. We will add time series were appropriate to highlight our case/analysis.

5. More detailed information of in-situ sites could be given or summarized.

   Response: We agree with this.

   Action: A table with more detailed information about the in-situ data locations will be given, including the data time period, lon/lat/altitude and climate zone. This will  coincide with a more detailed discussion of the result in terms of geographic areas and land cover types as requested by the other anonymous reviewer.

6. Figure 2, 3, 4 and A1, A2, A3 could also give the bar chart distribution.

   Response: Thanks, we agree again.

   Action: We will provide this information in the revised manuscript.

7. Please explain the reasons for case selections. e.g. 30 June 2018 in Figure 4 and 30 Sep 2018 in Figure A2.

   Response: There was no specific reason for the case selections. They are representative for other time steps.

   Action: This will be explicitly mentioned in the revised manuscript. Moreover, to enhance the representativeness of the selected cases, one of the examples will be changed to a day in winter.

8. If the LST and SNR products are compared with any other reanalysis or satellite products?

   Response: Yes, this could be included.

   Action: We will explore the possibility to include a comparison to a suitable state-of-the-art dataset, e.g., ERA5-Land.

---

## Author Comment (AC2)

The manuscript "High-resolution all-sky land surface temperature and net radiation over Europe" has been reviewed. The authors presented a methodology to combine the advantages of geostationary observations at high temporal resolution with observations from polar-orbiting satellites at high spatial resolution, resulting in a gap-free all-sky LST and net radiation dataset at 1-km spatial resolution and daily frequencies for 2018-2019 across Europe. This dataset is important for hydrological modelling and as input to models dedicated to estimating evaporation and surface turbulent heat fluxes. However, more comprehensive analysis on this dataset is required before further consideration.

We thank reviewer #2 for the time and effort put into reviewing our manuscript which, we believe, will result in a marked improvement of this study. We are happy to share our point-by-point responses highlighted in blue. In general, we agree that a more comprehensive analysis is required and will expand on the highlighted sections.

1. Lines 223-225. As the dataset includes all-sky land surface temperature, I think it is necessary to implement accuracy assessment to tell us the uncertainties of the produced LST data.

   Response: We agree with the reviewer.

   Action: We will expand on the LST validation add attempt to validate LST directly. The issue so far has been the lack of access to LST ground truth data and therefore we performed the validation for the resulting radiation datasets. We generally will include more validation statistics for the generated products, also per land cover type and seasonally, as also reviewer #1 commented on expanding the validation.

2. A discussion section is required to explain the results and to compare against existing datasets. For example, lines 217-218, why there are worse accuracy in Belgium for *SWin* and around the Alps for *Lwin*?

   Response: We agree with this. Both SWin and LWin are not produced in this study but are input products obtained from LSAF. Nevertheless, more context can be provided and will be added to the manuscript: It is fair to consider that the temporal variability of cloud cover determines to a large extent the variability of SW and LW. Furthermore, that is also the main information provided by satellite data (clouds and cloud optical depth via top-of-atmosphere reflectances). So the generally high R values for both SW and LW corroborate that satellite products follow reasonably well the in situ time-series.
   LW estimates require screen variables (LW is more indirectly linked with top-of-atmosphere observations than SW), which are derived from NWP - therefore it is not surprising that R and RMSE are not as good as those for SW. The accuracy of screen variables may also explain the worse performances of LW in the Alps - although some orographic corrections are performed, the uncertainty is likely larger in mountainous regions.

   Action: We will expand on the discussion of the results along these lines, including an accuracy assessment of these products in the respective literature as well as an analysis of

the time series at the mentioned locations with worse performance. This again is also in line with reviewer #1 asking for a more extensive validation.

3. Pearson's correlation coefficient and RMSE are not enough for validation. Examples of comparison of temporal patterns between estimated values and in-situ observations at typical stations are suggested. Meanwhile, the impact factors on the estimated variables can also be analyzed. For example, how does the RMSE change across seasons? Do land cover types significantly affect the accuracy of estimated variables? How about the accuracies in areas with and without missing satellite observations?

Response: We again agree that the validation requires more detail.

Action: We will add time series plots to the manuscript and systematically discuss and show differences in performance across the seasons, geographic areas and land cover. This again is in line with above comments and comments from reviewer #1. More extensive validation and discussion is indeed required.

Preliminary analysis showed deteriorating performance throughout the winter months due to increased cloud and snow cover which prevents the retrieval of clear-sky LST. The resulting dataset thus relies more on modelled all-sky estimates for LST and the incoming radiation products.

The emissivity dataset used in the study also relies on clear-sky observations and days with no observations are estimated by linearly interpolating between available data. The uncertainty thus also increases during winter and in regions with more frequent cloud cover. While the availability of the clear-sky estimates varies throughout the seasons there are no areas with no data at all.

We will also look into the uncertainty of in situ obs and representativeness issues, e.g., in cases when the station is far higher/lower than the pixel's average height).

4. Lines 42-57. A comprehensive summary of existing studies/datasets (including advantages and drawbacks) may help to emphasize the novelty of this study.

Response: We agree fully with this statement.

Action: We will expand this section and include a comprehensive overview of similar products and novel LST merging methods.

5. Lines 58-61. What research gaps have the authors solved? It is better to describe it here.

Response: The main research gap is the availability of high-resolution gap-free LST and net radiation datasets at at least daily resolution which can be either used for analysis or the forcing of hydrological/land  surface models. This is addressed by developing a  suitable

methodology to a) combine LST estimates from polar-orbiting and geostationary satellites in order to combine their advantages in temporal and spatial coverage, and b) combine the resulting merged LST product with other datasets to obtain a novel net radiation dataset.

Action: We will include this information in Sect 1 at the proposed position.

6. Lines 108-111. What is the overpass time for clear-sky LST estimates from Sentinel 3A and 3B, respectively? Why do the authors only use the data from Sentinel 3A.

Response: For this initial study focusing on 2018-2019 only Sentinel 3A data was used. Sentinel 3B was launched in April 2018 and was flown in tandem with Sentinel 3A from June to October of the same year after which it was moved to its nominal orbit, see e.g. https://www.mdpi.com/2072-4292/12/17/2668. The local overpass time of Sentinel 3A and Sentinel 3B thereafter is the same (ca. 10:30 am/pm) with the precise time depending on the latitude and taken into account in the merging methodology.

Action: When expanding the merged LST and net radiation dataset to more recent years, Sentinel 3B data will be included making the merged product more robust. However, in this pilot study focused on the 2018–2019 period we would like to focus on Sentinel 3A only.

7. Section 3.3. The performance of the merging method needs to be evaluated.

Response: Some of the main benefits are gained through the bias-correction steps (1-3). The subsequent assimilation step is to obtain a more "Sentinel-like" product with a more marginal impact.

Response: We will include more validation statistics and evaluation criteria, in line with previous comments of expanding the validation and also in response to reviewer #1.

8. Line 199. More details on the Kalman Filter can be added to make an easier understanding by readers.

Response: We agree.

Action: More detail will be added and when necessary, suitable references to the Appendix with a more in-depth description of the assimilation step will be added.

---

## Author Response (AR1)

**Author's response: High-resolution all-sky land surface temperature and net radiation over Europe**

Firstly, we thank the reviewers again for their very helpful feedback on improving the manuscript. We have taken into account their advice and adjusted the manuscript accordingly.

We'd also like to sincerely apologise for the delay in this review process.

Main changes to manuscript:
Most notably, both reviewers agreed on the fact that they were missing validation statistics, e.g. seasonal and for different land cover types. We have added this to the manuscript. Additionally, we have added a comparison between the here presented dataset and ERA5-Land as well as a table listing the in-situ stations used for the validation.

Below we list the point by point responses to the reviewer comments as well as changes which we have implemented. A few of the responses have been modified compared to the original ones when working on the revision of the manuscript.

**Reviewer 1:**

1. the high resolution LST product is merged from LEAF (all sky) and Sentinel 3 LST (clear sky). The two LSTs have different spatial and temporal resolutions. While doing the merging, if any cloud effect is considered? If any cloud product is involved? If yes, please indicated it.

   Response:

   We have not included any cloud product although the prevalence of clouds is considered in the merging procedure. Cloud cover is assumed when no LSAF clear-sky data are available and then we fall back to the all-sky dataset. Therefore:

   1) Cloud over is taken into account by only using clear-sky LSAF data for the computation of the bias between LSAF and Sentinel-3, both daytime and nighttime.

   2) When computing the diurnal cycle effect on the misalignment of the Sentinel-3 observation to the full hour, this is only done if the two required LSAF samples on which the linear-interpolation is performed are clear-sky observations.

    3) Due to step 1 and 2, In the Kalman assimilation scheme a higher uncertainty is assumed for LSAF all-sky products than for the clear-sky equivalent.

   Action: This is described in the LST merging methodology section and we have added e.g. "The all-sky LSAF product, which contains modelled LST when cloud cover prevents the direct retrieval, enables the merged gap-free LST product with Sentinel-3 resolution." for further clarification.

In the conclusion we have added "Here, we presented a methodology to combine the advantages of geostationary LST and radiation observations, enhanced with modelled data when cloud cover inhibits the direct retrieval, at high temporal resolution, with observations from polar-orbiting satellites at high spatial resolution, resulting in a gap-free all-sky LST and net radiation dataset for 2018–2019 across Europe." for further clarification.

2. while downscaling the LST product, if any edge effects (coast lines, cloud edges) are considered?

   Response:

   Due to the coarser spatial resolution of LSAF coast lines are not as well represented as in the 1 km Sentinel-3 data, as pixels with predominant open water coverage are masked out. The merging procedure and calculation of all-sky LST and net radiation at 1 km resolution can only be carried out where all input datasets are available and therefore the same rugged coast lines are visible in the LST/net radiation product.

   Action: A simple solution, albeit introducing some uncertainties, is to extrapolate from the closest pixels with valid data on a daily basis. We have implemented this for data visualisation and will apply the same correction to the dataset prior to its public release.

3. Line 220, it is said "Extensive validation of the LSAF and Sentinel 3 LST products has already been performed (see below). Both have an average accuracy below 1.5 K, although it varies across space and time. Our goal is to combine their individual strengths in terms of spatial and temporal resolution to obtain an enhanced representation of landscape heterogeneity". Although there are extensive validations of the LSAF and Sentinel 3 LST products, the validations are based on different spatial and temporal resolutions. It does not mean that the merged product could also has a good performance. It is good to give the statistics.

   Response: We agree with the reviewer. However, we argue that the generated LST product is in essence an intermediate product for the final merging of the net radiation dataset as the final output. The validation is therefore conducted indirectly by validating outgoing longwave radiation at the available in-situ sites located throughout Europe. This is valid as LST is the only dataset that is modified, thus any changes relate to a change in LST.

   Action: For the outgoing radiation, as well as the other radiation components, we have added validation metrics as well as seasonal and land cover analysis. We have generally expanded on the validation making it more insightful. We hope that the reviewer is content with this option (see section 4 as well as appendix B, C, D.)

4. The paper is lack of statistics. e.g. figure 1, any overall statistics could be summarized in a table? And the absolute RMSEs are given in Figure 1. The percentage-wise is worth known. And so does the validations of outgoing raditaions and SNR. Please summarize the overall

statistics (R, bias, RMSE (including percentages)), degree of freedom) in tables.

Response: We agree that they should be included.

Action: We have significantly extended the validation by adding more performance metrics as well as seasonal and land cover analysis with figures showcasing this. We have added a table in the appendix containing the overall and seasonal validation statistics.

5. More detailed information of in-situ sites could be given or summarized.

Response: We agree.

Action: We have added the table listing the available in-situ sites (see Apendix A).

6. Figure 2, 3, 4 and A1, A2, A3 could also give the bar chart distribution.

We generally agree that this could be useful but given that we have added many figures to the new version of the manuscript we would like to skip this, if the reviewer agrees. Otherwise we would be happy to add this in the final version.

7. Please explain the reasons for case selections. e.g. 30 June 2018 in Figure 4 and 30 Sep 2018 in Figure A2.

Response: There was no specific reason for the case selections. They are representative for other time steps.

Action: This is now clearly mentioned "This day was chosen for no particular reason and is representative for other dates."

8. If the LST and SNR products are compared with any other reanalysis or satellite products?

Response: Yes, good point.
Action: We have added a comparison of the final daily net radiation product to ERA5-Land.

**Reviewer 2:**

1. Lines 223-225. As the dataset includes all-sky land surface temperature, I think it is necessary to implement accuracy assessment to tell us the uncertainties of the produced LST data.

Response: The issue is that there is a lack of access to LST validation sites. Given that the purpose of our LST is to be an intermediate product for the calculation of outgoing longwave radiation we have focused our attention on the latter. We argue that, for the intent of this specific study, this is more useful.

Action: The validation of all radiation components has been extended with more validation metrics as well as validation per season and land cover type. The only dataset which is modified for outgoing longwave radiation is LST, so any changes directly relate to this.

2. A discussion section is required to explain the results and to compare against existing datasets. For example, lines 217-218, why there are worse accuracy in Belgium for *SWin* and around the Alps for *Lwin*?

Response: Both SWin and LWin are not produced in this study but are input products obtained from LSAF and well validated in the literature. Nevertheless, we agree and more context could be provided.

Action: This text has been added to the manuscript:

"It is fair to consider that the temporal variability of cloud cover determines to a large extent the variability of SW and LW. Furthermore, that is also the main information provided by satellite data (clouds and cloud optical depth via top-of-atmosphere reflectances). So the generally high R values for both SW and LW corroborate that satellite products follow reasonably well the in situ time series.
LW estimates require screen variables (LW is more indirectly linked with top-of-atmosphere observations than SW), which are derived from numerical weather prediction models – therefore it is not surprising that R and RMSE are not as good as those for SW. The accuracy of screen variables may also explain the worse performances of LW in the Alps; although some orographic corrections are performed, the uncertainty is likely larger in mountainous regions."

3. Pearson's correlation coefficient and RMSE are not enough for validation. Examples of comparison of temporal patterns between estimated values and in-situ observations at typical stations are suggested. Meanwhile, the impact factors on the estimated variables can also be analyzed. For example, how does the RMSE change across seasons? Do land cover types significantly affect the accuracy of estimated variables? How about the accuracies in areas with and without missing satellite observations?

Response: We agree that the validation requires more detail.

Preliminary analysis showed deteriorating performance throughout the winter months due to increased cloud and snow cover which prevents the retrieval of clear-sky LST. The resulting dataset thus relies more on modelled all-sky estimates for LST and the incoming radiation products.

The emissivity dataset used in the study also relies on clear-sky observations and days with no observations are estimated by linearly interpolating between available data. The uncertainty thus also increases during winter and in regions with more frequent cloud cover. While the availability of the clear-sky estimates varies throughout the seasons there are no areas with no data at all. We have added some time series plots to the manuscript.

Action: We have added additional performance metrics and systematically discuss and show differences in performance across the seasons, geographic areas and land cover. This again is in line with above comments and comments from reviewer #1.

4. Lines 42-57. A comprehensive summary of existing studies/datasets (including advantages and drawbacks) may help to emphasize the novelty of this study.

   Response: Thank you for the suggestion. We agree this would be interesting, but this could perhaps be for a systematic review paper. If the reviewer disagrees, we are happy to add some more examples.

5. Lines 58-61. What research gaps have the authors solved? It is better to describe it here.

   Response: The main research gap is the availability of high-resolution gap-free LST and net radiation datasets at at least daily resolution which can be either used for analysis or the forcing of hydrological/land surface models. This is addressed by developing a suitable and generic methodology to a) combine LST estimates from polar-orbiting and geostationary satellites in order to combine their advantages in temporal and spatial coverage, and b) combine the resulting merged LST product with other datasets to obtain a novel net radiation dataset.

   Action: We have modified the introduction to better describe the rationale of this study, e.g. "The novelty of this study lies in systematically exploiting the advantages, and mitigating the disadvantages, in terms of spatial and temporal resolution of available observations, which are well validated, in a physical and consistent manner and assembling a net radiation dataset based on the individual incoming and outgoing radiation components."

6. Lines 108-111. What is the overpass time for clear-sky LST estimates from Sentinel 3A and 3B, respectively? Why do the authors only use the data from Sentinel 3A.

   Response: For this initial study focusing on 2018–2019 only Sentinel 3A data was used. Sentinel 3B was launched in April 2018 and was flown in tandem with Sentinel 3A from June to October of the same year after which it was moved to its nominal orbit, see e.g. https://www.mdpi.com/2072-4292/12/17/2668. The local overpass time of Sentinel 3A and Sentinel 3B thereafter is the same (ca. 10:30 am/pm) with the precise time depending on the latitude and taken into account in the merging methodology.

   Action: When expanding the merged LST and net radiation dataset to more recent years, Sentinel 3B data will be included making the merged product more robust. However, in this pilot study focused on the 2018–2019 period we would like to focus on Sentinel 3A only. We have clarified the use of Sentinel-3A and the above information in the manuscript. [quote...]

7. Section 3.3. The performance of the merging method needs to be evaluated.

   Response: Some of the main benefits are gained through the bias-correction steps (1-3). The

subsequent assimilation step is to obtain a more "Sentinel-like" product, and has a more marginal impact. Generally, as the LST product is essentially a useful intermediate product to obtain SNR, and LST in situ measurements are limited, we have validated outgoing longwave measurements instead..

Action: We have included more validation statistics and evaluation criteria, in line with previous comments of expanding the validation and also in response to reviewer #1.We have also clarified the approach taken in the validation of LST, for instance in the discussion section: "It is to be noted that while a gap-free LST datset was developed within this study, the validation of the dataset was carried out   indirectly based on LWout measurements. This served the purpose of the study to ultimately create a SNR dataset."

8. Line 199. More details on the Kalman Filter can be added to make an easier understanding by readers.

Response: Thank you for the suggestion

Action: We have modified the description in the Annex F to make it clearer.

---

## Referee Report (RR1)

1. The novelty of the proposed algorithm should be emphasized. Motivation is the key to the introduction section. Need improvement: Summarize the knowledge gap here and justify why a new approach is needed.

2. P3, it is recommended to provide a summary of previous methods used for generating all-weather LST and SNR in the introduction part. For instance, mention relevant papers such as:

Jia, Aolin, et al. "Global hourly, 5 km, all-sky land surface temperature data from 2011 to 2021 based on integrating geostationary and polar-orbiting satellite data." Earth System Science Data 15.2 (2023): 869-895.

Xu, Shuo, and Jie Cheng. "A new land surface temperature fusion strategy based on cumulative distribution function matching and multiresolution Kalman filtering." Remote Sensing of Environment 254 (2021): 112256.

3. The data quality of in-situ measurements was not well displayed, which is crucial to evaluate the reliability of satellite data. It is recommended to add more detailed information such as the instruments used and the accuracy of station observations.

4. Line 240. The worse match between observations and in situ data may indicate high spatial heterogeneity. If the site is located in an area with high spatial heterogeneity, it may not be suitable for validating satellite data. It is important to provide additional clarification regarding the factors contributing to the worse validation results observed at certain sites.

5. Section 4.2. When discussing the merging of LST, it is important to compare the merged LST with in-situ measurements. Include a comparative analysis between the merged LST data and the corresponding in-situ measurements to demonstrate the accuracy and reliability of the merging process.

6. During the validation process, it is crucial to report the accuracy of clear-sky data (such as LST and SNR) separately from the accuracy of cloudy-sky data. This differentiation is important as it provides a comprehensive evaluation of the algorithm's performance under varying sky conditions.

7. In the conclusion section, it is important to outline the novel aspects and improvements introduced by the proposed method and the generated products, highlighting their advancements compared to existing methods and published products.

---

## Referee Report (RR2)

The revised manuscript "High-resolution all-sky land surface temperature and net radiation over Europe" has been largely improved. However, there are still several concerns from my viewpoint.

1. In the response to my comment 1, authors explained that the land surface temperature (LST) data is an intermediate product and did not need accuracy assessment. However, the LST data has been shown in the title of the manuscript and described in the abstract. As a published dataset, the LST data should be reliable to be used. Otherwise, no need to be shown in the title.

2. In the response to my comment 4, authors think that a systematic review of existing LST datasets in not necessary. The ESSD journal focus on the novelty and description of the published datasets instead of the novelty of the method. Therefore, a solid description on existing datasets is very important to show the novelty of this study. I suggest authors highlight the novelty of their dataset by comparing with existing datasets. As I know, there are several other gap-less LST datasets covering the current study area (even larger areas). No needs for listing all the literature, but it is necessary to do discussion).

3. The authors are suggested to double check their descriptions. Examples of issues are listed as follows. 1) Lines 70 and 84, what does "??" mean? 2) Line 224, what does "nore details" mean? 3) Line 378, what does "the the Sentinel 3 LST…" mean?

---

## Author Response (AR2)

We thank the reviewers of this manuscript. The constructive criticism has helped to further improve it and clarify some remaining issues. We provide below a point-by-point response to each of the reviewers comments, highlighted in green.

We'd like to mention that the dataset has been updated in accordance with a previous reviewer comment. Both the land surface temperature and net radiation dataset are identical to the previous version but have been post-processed to better follow coast-lines and inland water bodies.

Most importantly, as we understand, is that both reviewers have taken an issue with the lack of direct validation of the land surface temperature dataset. We had argued that the validation based on outgoing longwave radiation against FluxNet measurements should be sufficient. We understand this remains an issue and have followed the comments of Reviewer #2 and have reworded the title from "High-resolution all-sky land surface temperature and net radiation over Europe" to **"1 km all-sky net radiation over Europe enabled by the merging of land surface temperature retrievals from geostationary and polar-orbiting satellites."**

**Reviewer #1 response**

1. The novelty of the proposed algorithm should be emphasized. Motivation is the key to the introduction section. Need improvement: Summarize the knowledge gap here and justify why a new approach is needed.

Thank you for the suggestion. We have expanded the motivation of the study in the introduction section highlighting that our approach is consistently based on the surface energy balance by downscaling individual radiation components. This seems to be the most 'natural' way of producing high resolution net radiation estimates from the available data sources.

2. P3, it is recommended to provide a summary of previous methods used for generating all-weather LST and SNR in the introduction part. For instance, mention relevant papers such as:
Jia, Aolin, et al. "Global hourly, 5 km, all-sky land surface temperature data from 2011 to 2021 based on integrating geostationary and polar-orbiting satellite data." Earth System Science Data 15.2 (2023): 869-895.

Xu, Shuo, and Jie Cheng. "A new land surface temperature fusion strategy based on cumulative distribution function matching and multiresolution Kalman filtering." Remote Sensing of Environment 254 (2021): 112256.

We now mention existing studies in the introduction as well as Discussion, including the ones mentioned above. In addition mentioning the papers we shortly describe the approach/methodology they present.

3. The data quality of in-situ measurements was not well displayed, which is crucial to evaluate the reliability of satellite data. It is recommended to add more detailed information such as the instruments used and the accuracy of station observations.

We acknowledge that the FLUXNET measurements have an error (difference to the 'truth' at the local scale that they sample), but that the pixel-to-local 'representativeness' error (difference between pixel truth that we aim for, and the local truth the smaller tower footprint) is much larger.

Unfortunately, we cannot solve this issue but argue that using as many stations as possible benefits the validation, also in areas where the spatial heterogeneity is large (see also next comment).

We have added this statement to the Discussion section, see page 21, L386.

4. Line 240. The worse match between observations and in situ data may indicate high spatial heterogeneity. If the site is located in an area with high spatial heterogeneity, it may not be suitable for validating satellite data. It is important to provide additional clarification regarding the factors contributing to the worse validation results observed at certain sites.

Thank you, we agree with the observation. We have clarified this in the next. We also argue the case for keeping all available in-situ data as the availability is already quite sparse and we want to carry out the validation in as many locations as possible, even under challenging circumstances.

"Since the availability of in-situ measurements is already fairly limited, we argue that carrying out the validation also in challenging terrain benefits the overall accuracy assessment." P9, L258.

5. Section 4.2. When discussing the merging of LST, it is important to compare the merged LST with in-situ measurements. Include a comparative analysis between the merged LST data and the corresponding in-situ measurements to demonstrate the accuracy and reliability of the merging process.

As highlighted at the very top of this reviewer response, we have modified the title. This change accounts for net radiation being the focus and the final product of this study. Land surface temperature has been indirectly validated via longwave outgoing radiation against FluxNet measurements but we agree that land surface temperature should not be presented as a final validated dataset in the title. It is a means to the end. We have also modified other parts of the manuscript to reflect this shift of focus, e.g. the abstract and introduction.

6. During the validation process, it is crucial to report the accuracy of clear-sky data (such as LST and SNR) separately from the accuracy of cloudy-sky data. This differentiation is important as it provides a comprehensive evaluation of the algorithm's performance under varying sky conditions.

We have added a comparison of the final SNR product in terms of performance between clear-sky and cloudy sky days in section 4.5. See Page 17 text and Figure 8.

7. In the conclusion section, it is important to outline the novel aspects and improvements introduced by the proposed method and the generated products, highlighting their advancements compared to existing methods and published products.

We have added existing studies to the Conclusion section of the paper. Conceptually we highlight their differences. A quantitative comparison has not been added as this is not the scope of this paper and would prove difficult due to differing spatial and temporal domains.

**Reviewer #2 response**

The revised manuscript "High-resolution all-sky land surface temperature and net radiation over Europe" has been largely improved. However, there are still several concerns from my viewpoint.

1. In the response to my comment 1, authors explained that the land surface temperature (LST) data is an intermediate product and did not need accuracy assessment. However, the LST data has been shown in the title of the manuscript and described in the abstract. As a published dataset, the LST data should be reliable to be used. Otherwise, no need to be shown in the title.

We understand the concerns of the reviewer and have directly addressed this by modifying the title. The focus of this study is on producing net radiation data and the land surface temperature merging served this overall goal. While the temperature was indirectly validated via outgoing longwave radiation against FLUXNET measurements a direct validation would be necessary when directly advertising a novel land surface temperature product.

2. In the response to my comment 4, authors think that a systematic review of existing LST datasets in not necessary. The ESSD journal focus on the novelty and description of the published datasets instead of the novelty of the method. Therefore, a solid description on existing datasets is very important to show the novelty of this study. I suggest authors highlight the novelty of their dataset by comparing with existing datasets. As I know, there are several other gap-less LST datasets covering the current study area (even larger areas). No needs for listing all the literature, but it is necessary to do discussion).

We thank the reviewer for their comment and have now added several recent studies focusing on the creation of all-sky land surface temperature products. In the newly added Discussion section, see page 20, we mention the methodologies and how our approach differs. This is partly also handled in the introduction section.

3. The authors are suggested to double check their descriptions. Examples of issues are listed as follows. 1) Lines 70 and 84, what does "??" mean? 2) Line 224, what does "nore details" mean? 3) Line 378, what does "the the Sentinel 3 LST…" mean?

Thank you very much. We have carefully revised the manuscript to avoid these mistakes.

---

## Author Response (AR3)

RC1:

The manuscript has improved, but I still have some concerns.

In response to my comment #3, the authors suggest that FLUXNET measurements do have errors. Therefore, I would suggest adding information about the observational instrumentation, such as the sensors and their accuracy. I also suggested adding a description of the sources of error in the field data.

**We have added a section describing the measurement technique and to-be-expected uncertainties, see Line 164-171:**

*While the in situ measurements are considered as ground-truth, it is necessary to mention that they have their own sources of uncertainties. Incoming shortwave and longwave radiation are measured by pyranometers and pyrgeometers. Accuracy targets for the BSRN network measurements (from 2004) are for example 2% or 5 W m$^{-2}$ for incoming shortwave radiation and 2% or 3 W m$^{-2}$ for incoming longwave radiation. Target uncertainties for outgoing shortwave and longwave radiation are 3% and 2% (or 3 W m$^{-2}$) respectively (McArthur, 2004). For the measurement of the outgoing radiation components the pyranometer/pyrgeometer is installed facing downwards. The target uncertainties are in line with the achievable accuracy of the pyranometer/pyrgeometer instruments although they might not be met under some conditions, e.g. incorrect installation at an angle or snow cover. The instruments should be calibrated every 2 years (Walter-Shea et al., 2019).*

In addition, the authors stated that 'using as many stations as possible benefits the validation, also in areas where the spatial heterogeneity is large'. However, the use of in situ data from regions with large spatial heterogeneity may lead to inaccurate or erroneous validation results.

**We thank the reviewer for this comment and understand the concern. The high variability expected in heterogeneous landscapes in the end determines the need for more *in situ* measurements within the satellite footprint to be able to reduce representativeness errors and make the validation of 1 km data more meaningful. Ideally, we would have many validation sites in every heterogeneous pixel to capture the entire footprint. As unfortunately this is currently not possible, we argue that using as many sites as possible, at least to some extent, alleviates the issue as in theory the representativeness errors will average out the more *in situ* data across the domain is used. While this approach is not perfect, restricting the validation to homogeneous pixels would further reduce the limited amount of *in situ* data and make the validation less meaningful.**

**We rephrased L434–435 as:**

*Unfortunately, we cannot solve this issue but argue that using as many stations as possible benefits the validation, underline(particularly within pixels) where the spatial heterogeneity is very large.*

RC2:

The manuscript has been further improved but is still not satisfactory. There are several issues in the revised manuscript. Authors can consider to further improve this manuscript if they can clearly proof the quality and novelty of their datasets. Issues I can find as follows.

1. "1 km" in the title is not proper since the first word is usually not number. Move "1 km" to the middle of the title or use "One-kilometer" may be better.

**We thank the reviewer for this comment and agree. We have changed the title to** *"High-resolution (1-km) all-sky net radiation over Europe enabled by the merging of land surface temperature retrievals from geostationary and polar-orbiting satellites."*

2. If I remember correctly, the links on the published datasets should also be shown in the abstract.

**Thank you, we have added the links to the abstract (L17-18).**

3. Lines 349-369. As the LST data is not the main output of this study, it is not necessary to compare it with other existing LST datasets. In my previous comment, "I suggest authors highlight the novelty of their dataset by comparing with existing datasets". It is meaningful to compare SNR datasets instead of LST datasets. Besides, the current comparisons of LST datasets are unsatisfactory as the advantages and drawbacks of these LST datasets are still not clear to us.

**We thank the reviewer for this comment and agree. Important other studies were listed but an evaluation in terms of potential advantages or drawbacks was not included. We have added this now for the LST dataset and have also expanded the comparison for SNR datasets (L359–424):**

[revised manuscript text omitted]
 have also compared the SNR dataset to the state-of-the-art ERA5-Land reanalysis product, see Figure 11 and 12 and Lines 330-350:**

*Figure 11 shows the SNR validation for the different CCI land cover types for a LSAF only based SNR as well as the downscaled product. The Figure also includes performance metrics for the ERA5-Land product (Muñoz-Sabater et al., 2021) which were included to give some context. R is generally high for all products (ca. 0.95) for all sites with the exception of sites with land cover affected by water. There ERA5-Land outperforms the LSAF and downscaled SNR product in terms of R, likely due to a sub-optimal treatment of these areas in the processing of the input products. In terms of MSE ERA5-Land again outperforms the other products for water affected land cover. However, for the other land cover classes the LSAF SNR and downscaled products perform better with the downscaled dataset showing the lowest values. In terms of bias, ERA5-Land performs best with the downscaled data performing between ERA5 and the LSAF only SNR. Figure 11. Validation of SNR for different CCI land cover types in terms of R, RMSE, RMSPE and bias.*

*For the SNR products we also carry out a seasonal analysis. The results of this are shown in Figure D1 and Figure D2 in boxplot form (see annex). Table E1 and Table E2 list all performance metrics for the entire study period as well as seasonally. For the entire 2018–2019 period, R is very similar for both datasets with R=0.93 for the downscaled product and R=0.92 for ERA5-Land. In comparison to ERA5-Land, the downscaled product has a RMSE of 22.53 vs 25.7 W 2. The average bias is lower for ERA5-Land, with -1.56 vs -6.83 W 2.*

*The downscaled product shows a better performance for the summer period AMJ and JAS (R=0.91 and 0.93 vs 0.83 and 0.86) and the same is true in terms of RMSE (27.58 and 22.18 W 2 vs 34.79, 29.37 W 2). The seasonal bias is lower for the downscaled product. Figure 12 shows as an example the SNR for the downscaled product and ERA5-Land for the 30th of June over an area of western Europe. The increase in spatial resolution and therefore landscape details is clearly visible. The downscaled dataset both shows higher and lower values than ERA5-Land as it is able to resolve finer land surface features due to the high-resolution merged LST and Albedo inputs.*

4. Lines 371-390. The comparisons of methods is not satisfactory as the advantages of the proposed method is not persuasive enough.

**We have highlighted some points which we think are an important advantage of the proposed methodology. See reviewer point 3 above where we describe the amendments to the discussion section, both for the LST and SNR dataset. There we also refer to the quantitative comparison of the SNR product to ERA5-Land.**

5. Figure 11. Cannot find RMSE and RMSPE in this figure, but they occur in the title. The legend of the third subfigure may be not correct. Similar issues may exist in other figures.

**Thank you, this has been corrected and the manuscript has been proof-read.**